# The influence of idealized surface heterogeneity on virtual turbulent flux measurements

Frederik De Roo[1] and Matthias Mauder[1,2]

[1] Institute of Meteorology and Climate Research, Atmospheric Environmental Research (IMK-IFU), Karlsruhe Institute of Technology (KIT), Kreuzeckbahnstrasse 19, 82467 Garmisch-Partenkirchen, Germany
[2] Institute of Geography and Geoecology (IfGG), Karlsruhe Institute of Technology (KIT), Kaiserstrasse 12, 76131 Karlsruhe, Germany

*Correspondence to:* Frederik De Roo (frederik.deroo@kit.edu)

**Abstract.**

The imbalance of the surface energy budget in eddy-covariance measurements is still an unsolved problem. A possible cause is the presence of land surface heterogeneity, which affects the boundary-layer turbulence. To investigate the impact of surface variables on the partitioning of the energy budget of flux measurements in the surface layer under convective conditions, we set up a systematic parameter study by means of large-eddy simulation. For the study we use a virtual control volume approach, which allows the determination of advection by the mean flow, flux-divergence and storage terms of the energy budget at the virtual measurement site, in addition to the standard turbulent flux. We focus on the heterogeneity of the surface fluxes and keep the topography flat. The surface fluxes vary locally in intensity and these patches have different length scales. Intensity and length scales can vary for the two horizontal dimensions but follow an idealized chessboard pattern. Our main focus lies on surface heterogeneity of the kilometer scale, and one order of magnitude smaller. For these two length scales, we investigate the average response of the fluxes at a number of virtual towers, when varying the heterogeneity length within the length scale and when varying the contrast between the different patches. For each simulation, virtual measurement towers were positioned at functionally different positions (e.g. downdraft region, updraft region, at border between domains, etc.). As the storage term is always small, the non-closure is given by the sum of the advection by the mean flow and the flux-divergence. Remarkably, the missing flux can be described by either the advection by the mean flow or the flux-divergence separately, because the latter two have a high correlation with each other. For kilometer scale heterogeneity, we notice a clear dependence of the updrafts and downdrafts on the surface heterogeneity, and likewise, we also see a dependence of the energy partitioning on the tower location. For the hectometer scale we do not notice such a clear dependence. Finally, we seek correlators for the energy balance ratio in the simulations. The correlation with the friction velocity is less pronounced than previously found, but this is likely due to our concentration on effectively strongly to freely convective conditions.

# 1 Introduction

## 1.1 The role of landscape heterogeneity in the energy balance closure problem

The interpretation of the turbulent fluxes of latent and sensible heat at the Earth's surface still suffers from the unresolved energy balance closure problem of the eddy covariance (EC) measurement technique. That is, the measured turbulent fluxes do not close the available energy at the earth's surface (e.g., Foken, 2008; Leuning et al., 2012). There is an ongoing debate whether the missing energy can perhaps be solely described by additional missing terms related to energy conversion and storage or that the imbalance is a consequence of measurement errors in the velocity measurement due to flow distortion from the sonic anemometer pins. With respect to flow distortion, Horst et al. (2015) quoted an error of maximal 5%, but Kochendorfer et al. (2012) and Frank et al. (2013) claimed an error up to 15%. In response to the 15% error, one of us (Mauder, 2013) has pointed out some counterevidence, and a recent modeling study by Huq et al. (2017) on flow distortion did not find evidence for such large errors either. In short, it is unlikely that the previously mentioned issues can explain the fact that very different sites around the world often exhibit an imbalance of more than 20% (e.g., Wilson et al., 2002; Hendricks-Franssen et al., 2010; Stoy et al., 2013).

In fact, the studies by Mauder et al. (2007) and Stoy et al. (2013) have shown that a common property among sites that do not close the energy balance, is a more pronounced surface heterogeneity on the landscape-scale. This motivates us to investigate the energy balance closure problem in the context of landscape heterogeneity. Moreover, Stoy et al. (2013) also found a good correlation between the friction velocity ($u_*$) and the energy balance closure. This result was reproduced by Eder et al. (2015b) by means of a study combining Doppler wind LiDAR and EC tower data. The same correlation has also been noticed in a recent year-long large-eddy simulation (LES) by Schalkwijk et al. (2016) and in an idealized LES study by Inagaki et al. (2006). In addition, the study of Eder et al. (2015b) could relate the energy balance residual to the mean gradients in the lower boundary-layer, thereby providing more evidence for the connection between the energy imbalance and the presence of quasi-stationary structures in the boundary layer. These circulations typically arise in heterogeneous terrain but may also develop over a completely homogeneous surface to a lesser extent, depending on the atmospheric stability regime, due to self-organization. Persistent updrafts and downdrafts tied to the landscape heterogeneity have been found e.g. by Mauder et al. (2010) during the 2008 Ottawa field campaign. In the case of cellular convection in heterogeneous terrain the distinction between the primary and the secondary circulation becomes blurred, when the convection cells are tied to the landscape heterogeneity.

## 1.2 The influence of landscape heterogeneity on the boundary-layer structure

The influence of heterogeneous landscapes on properties of the atmospheric boundary-layer has already been investigated for a few decades with numerical models, primarily large-eddy simulation. We will summarize a few results that are relevant to the non-closure of the energy balance. Avissar and Chen (1993) obtained significant mesoscale fluxes tied to the terrain heterogeneity. These mesoscale fluxes are carried by the vertical wind of the meso-scale circulations, however, they are not present at the ground level. Raupach and Finnigan (1995) also found that surface heterogeneity induces boundary-layer motions, nevertheless the area-averaged properties, including the fluxes, were not significantly influenced by the heterogeneity or

the circulation. At the first glance, both statements appear in conflict with a generic influence of the landscape heterogeneity around a measurement site on the energy balance closure.

On the other hand, Shen and Leclerc (1995) found that the horizontally averaged variances and covariances were influenced by land surface heterogeneity with scales smaller than the boundary-layer depth. This was also confirmed by Raasch and Harbusch (2001). This apparent contradiction can be explained by the fact the resolution of these models was coarse due to computational restrictions at that time, which has a few implications. Firstly, from continuity we indeed expect no vertical meso-scale transport by advection with the mean flow at the lowest grid point representing the lower surface, since w=0 due to the rigid no-slip boundary, but horizontal flux-divergence plays a role, too. Secondly, we should keep in mind that areally averaging over sufficiently large distances represents a form of spatial filtering due to the coarse resolution. Steinfeld et al. (2007) argued that a spatial filtering method will yield energy balance closure, whereas single-tower temporal averaging of the sensible heat flux signal in heterogeneous domain suffers from low-frequency contributions due to the shifted co-spectrum.

In summary, the previously mentioned studies showed that landscape hetereogeneity can induce mesoscale motions in the boundary-layer, especially for heterogeneity of length scales larger than the boundary-layer height. By using a large-eddy simulation model coupled to a land-surface-scheme, Patton et al. (2005) investigated striplike heterogeneities between 2 and 30 km. They found that the heterogeneities with length scales of 4 to 9 km were the most influential in altering the structure of the boundary-layer. A similar coupled model approach was used by Brunsell et al. (2011) to study three heterogeneity scales (approximately $10^{-1}z_i$, $z_i$, $10 z_i$, with $z_i$ the boundary-layer height). They found that only in the surface layer the length scale of the heterogeneity affected the spectral signature of the turbulent heat fluxes, and signals appeared blended in the mixed layer. Still, for the heterogeneity length of $10z_i$, secondary circulations arising from surface heterogeneity that extend through the whole boundary-layer were found. Furthermore, Brunsell et al. (2011) found that the partitioning between latent and sensible heat was affected by the scale of heterogeneity as the simulations for the intermediate scales led to a higher Bowen ratio. Since the intermediate scales (of scale $z_i$) appear more heterogeneous than the small or the large scales, this points toward the dominant influence of the sensible heat flux. Charuchittipan et al. (2014) also suggested to ascribe a larger fraction of the residual to the sensible heat flux than to the latent heat flux. The influence of synthetic surface heterogeneity on the Bowen ratio was also investigated by Friedrich et al. (2000) who found a non-linear response of the aggregated Bowen-ratio on the underlying land-surface distribution. Bünzli and Schmid (1998) investigated idealized heterogeneity by means of a two-dimensional $E - \epsilon$ model and found good correspondence with an analytical averaging scheme based on the context of a numerical blending height.

Although the above findings indicate that surface heterogeneity at scales of boundary-layer depth and larger can couple to the full boundary layer, surface heterogeneity at scales considerably smaller than the boundary-layer height appears to be blended, as observed by Raupach and Finnigan (1995). Furthermore, Avissar and Schmidt (1998) found that under a mild background wind, the influence of surface heterogeneity is quickly destroyed, in accordance with the findings of Hechtel et al. (1990). However, Maronga and Raasch (2013), who performed LES simulations for the response of the convective boundary layer in realistic heterogeneous terrain, advised that sufficient time and ensemble averaging is needed to extract the heterogeneity-induced signal, and they concluded that the upstream surface conditions can still influence the boundary-layer

properties under light winds. Albertson and Parlange (1999) showed that blending of the surface heterogeneity appears even under convective conditions, except for very large heterogeneities. However, Suehring and Raasch (2013) suggest that the blending of the surface follows from insufficient averaging. Therefore an apparent blending does not necessarily imply that small-scale surface heterogeneity could not have an influence on the energy budget at the surface. However, if smaller scales are

indeed completely blended in the mixed layer and therefore do not lead to circulations that involve the full boundary-layer, then we cannot expect non-surface layer properties (say, bulk gradients in the mixed layer or entrainment parameters) to correlate well with the energy balance residual. Though, even in the blended case small scale heterogeneity could still influence the surface energy budget through motions in the surface layer when the latter survive half-hour averaging. Indeed, for suburban terrain Schmid et al. (1990) noted significant differences in energy balance ratios at scales of $10^2 - 10^3$ m, presumably due to

micro-advection between the patches of different surface type.

## 1.3   Scope of this paper

Acknowledging the connection between the energy imbalance and quasi-stationary flow on the one hand, and quasi-stationary flow and surface heterogeneity on the other hand, we will investigate the effect of surface heterogeneity on the energy balance closure problem in this work. To this end, we will study a series of synthetic idealized landscapes that consist of a chess-board

pattern of surface fluxes with different amplitude and different wavelength in the $x$ and the $y$ direction. We will quantify the average influence on virtual tower data, and perform principal component analysis to link the energy balance ratio with surface characteristics, boundary-layer properties and turbulence statistics. To disentangle the influence of the surface heterogeneity from that of the meteorology, we will focus on a set-up of free convection without a synoptic wind (which will effectively lead to strongly to freely convective conditions diagnosed by the virtual towers). As hinted to in Brunsell et al. (2011), in heterogeneous

terrain the sensible heat flux appears more important for the imbalance at the intermediate length scales considered in their work, and we shall therefore focus on simulations that are practically dry (we have added a very small moisture flux). In addition, as both the lack of closure and the strength of the circulations are most pronounced for strongly convective conditions, we will likewise focus on (effectively) strongly unstable conditions to free convection, with the unstability parameter $-z/L$ ranging from 1 to 5000. The $-z/L$ is different from $\infty$ because the convective conditions lead to cellular circulation patterns,

which locally induce a friction velocity at the surface, and due to its positiveness, there will also be a horizontally averaged $u_*$ different from zero, as we derive the friction velocity from the kinematic momentum flux ($\tau_0/\rho$), in the same manner as how it is applied in standard eddy-covariance measurements (e.g. Kaimal and Finnigan, 1994):

$$u_*^2 = \tau_0/\rho = \left( \overline{u'w'}^2 + \overline{v'w'}^2 \right)^{1/2} . \tag{1}$$

This definition of friction velocity by the momentum flux is found in general fluid mechanics as well (e.g. Landau and Lifschitz

1959). However, only in homogeneous flow, the friction velocity makes sense as a scaling parameter in Monin-Obukhov similarity theory. Therefore, we want to stress that when the friction velocity is derived from the mean velocity gradient, this is only valid in homogeneous flow. For conditions of free convection in homogeneous terrain the friction velocity derived from the mean velocity is clearly zero (even though free convection flow is locally inhomogeneous). As we do not have homogeneous

flow in our study of heterogeneous terrain, we will make use of the momentum flux (1) to derive the friction velocity. From the perspective of the tower measurement, by eddy-covariance measurements alone it cannot be distinguished if a measured $u_*$ follows from the wind aloft or locally from the convection-driven circulation. In addition, the circulation locally leads to advective terms that can influence the energy balance closure: e.g. near an updraft there will be horizontal convergence in the flow field. Even in homogeneous terrain these advective terms can lead to a non-closure of the surface energy budget (e.g. Kanda et al., 2004). Despite the issues related to blending, we will focus on heterogeneity of length scales between $10^2 - 10^3$ m since for these scales the energy imbalance is most pronounced. The intermediate scales of $O(10^3$ m$)$ are of the order of the boundary-layer depth under typical convective conditions for mid-latitude afternoons, whereas the smaller scales of $O(10^2$ m$)$ are of the order of the surface-layer height. To keep the terminology more general than typical convection for mid-latitude afternoons, we will refer to them as heterogeneity of kilometer scale and hectometer scale, however. According to the classification of Orlanski (1975) these length scales are at the lower end of the meso-gamma-scale and at the upper end of the micro-alpha-scale, respectively.

Previous investigations with LES on the energy budget had been limited to more regular terrain with at least one homogeneous dimension, see the works of e.g. Kanda et al. (2004), Inagaki et al. (2006), Steinfeld et al. (2007), or Huang et al. (2008). Typically, the storage term was subtracted from the surface flux and only the vertical components of the energy balance were considered: i.e. the turbulent flux and a meso-scale flux (i.e. vertical advection) arising from turbulent organized structures (TOS) or heterogeneity-induced meso-scale motions (TMC). On the contrary, we will also analyze the contribution of the storage flux to the energy imbalance explicitly. Furthermore, the results presented there hold for the domain-averaged imbalance, and the method used is limited to heterogeneous terrain with at least one homogeneous dimension. In this work, however, we can extend the analysis of the energy budget to a full budget of the turbulent fluxes, by including additional terms stemming from horizontal advection by the mean flow. We take full account of all horizontal and vertical energy balance components with a so-called control volume approach, as in Finnigan et al. (2003), Wang (2010), and Eder et al. (2015a). As such, a study of two-dimensional heterogeneous domains becomes possible.

Let us stress again the research questions of this paper. The first aim is to investigate the average influence on virtual flux measurements of land surface heterogeneity in the form of a variable surface heat flux, for a given length scale of the heterogeneity. We focus on length scales of the order of kilometer, and also on length scales of the order of hectometers. The second aim is to correlate the simulated energy balance ratio to various observables that can be obtained from the simulation output and that are also measurable in a realistic setting.

## 2 Methods

### 2.1 Simulation set-up

For our simulations we have made use of the LES model PALM (Maronga et al., 2015). More precisely, we ran our simulations with PALM version 3.9. PALM resolves the turbulence down to the scale of the grid spacing, all turbulence below is parameterized by implicit filtering. The closure model in PALM is a so-called 1.5-order closure scheme, where the equations for the

resolved velocities and scalars are derived by implicit filtering over each grid box of the turbulent Navier-Stokes equations, and where an additional prognostic equation for the turbulent kinetic energy is solved. The turbulent kinetic energy in PALM (the sum of the variance of the subgrid-scale velocities) allows to model the energetic content of the subgrid-scale motions, and because it is related to spatial filtering it should not be confused with the typical turbulent kinetic energy in eddy-covariance

measurements related to the averaging of a time series. Of course, the latter can be approximated by the resolved kinetic energy in PALM plus the subgrid-scale turbulent kinetic energy. Finally, the Reynolds fluxes that appear in PALM's filtered equations (the spatial covariances of the subgrid-scale quantities) are parameterized by a flux-gradient approach involving the resolved gradient and a diffusivity coefficient that depends on the before-mentioned turbulent kinetic energy, the grid spacing and the height above the lower surface. However, at the first gridpoint above the surface, Monin-Obukhov similarity theory is applied

and therefore the turbulence there is completely parameterized. It is worth to note that the application of MOST at the first grid point in an LES, is done locally and based on the instantaneous velocity.

Relevant parameters of the simulation setup are summarized in Table 1, the grid spacing is $10\,\mathrm{m}$ in all three dimensions and the domain size is $6 \times 6$ square kilometers in the horizontal, and $2.4\,\mathrm{km}$ in the vertical. Demanding that the subgrid-scale flux does not exceed $1\,\%$ of the resolved flux, we will place our virtual flux measurements at $50\,\mathrm{m}$ height. The boundary conditions

of the simulations are periodic in the lateral dimensions. For the velocity we have Dirichlet conditions at the bottom (i.e. rigid no-slip conditions) with zero vertical and horizontal wind. At the top the horizontal velocity is commonly set to the geostrophic wind and the vertical velocity is zero. However, we have turned the geostrophic wind off (this is a homogeneous horizontal pressure gradient): $(u_g, v_g) = (0, 0)$. Nevertheless, due to the differences in surface heating, local pressure gradients will still develop. For potential temperature and humidity we have Neumann conditions at the lower boundary (given by the surface

fluxes) and also at the top boundary (where the flux is given by the lapse rate at initialization). The domain is initialized with constant profiles for the velocity (equal to the geostrophic wind for $x$ and $y$ and zero for the vertical velocity). The initial profiles are homogeneous in $x$ and $y$ and for potential temperature ($\theta$) it reads

$$\theta(z) = 300\,\mathrm{K} - 0.01\,\mathrm{K/m} \times (z - 1\,\mathrm{km}) \times \mathcal{H}(z - 1\,\mathrm{km}) \tag{2}$$

with $\mathcal{H}(\cdot)$ the Heaviside function. The top of the domain is situated within a stable inversion layer, which prevents that the

turbulence within the boundary-layer is influenced by the vertical domain size. In the lateral dimensions the domain is about 3 to 5 times the boundary-layer depth. For the vertical velocity we have added an a very small subsidence term (leading to a vertical pressure gradient in the equations) for heights above 1 km to counteract the destabilizing influence of the surface heat flux, with the subsidence velocity $w_s = -0.00003\,(z - 1\,\mathrm{km})\,\mathrm{s}^{-1}$ for all simulations. The data are extracted for four hours after two hours of spin-up time. For each hour a data point is collected by averaging over virtual measurements sampled at

every second. As our focus lies on the influence of the surface characteristics, we concentrate in the present study on the wind circulations purely generated by the surface heat flux, without complicating the analysis with additional synoptic drivers such as e.g. a geostrophic wind.

We ran two suites of simulations, one suite with 144 simulated cases focusing on surface heterogeneity of the kilometer scale (Table 2), and another suite with 144 simulated cases focusing on surface heterogeneity of the hectometer scale (Table 3).

The simulations are driven by a spatially variable surface sensible heat flux, the variation of which is controlled by a few parameters. More precisely, the surface sensible heat flux $H$ at each surface point $(x, y)$ is determined by

$$H(x,y) = (1 + A_x \mathbb{1}(x/L_x))(1 + A_y \mathbb{1}(y/L_y)) H_0, \tag{3}$$

where $\mathbb{1}$ is a antisymmetric periodic function with period equal to 2, and alternating between $-1$ and 1, i.e.

$$\mathbb{1}(x) = \sin(\pi x)/|\sin(\pi x)|. \tag{4}$$

The amplitudes of the two-dimensional surface heat flux is given by $A_x$ and $A_y$ and the periods by $L_x$ and $L_y$. $H_0$ is the average surface heat flux. In Fig. 2 we show an example of a synthetic surface heat flux as in (3) creating eight patches on the surface with four different values for the surface sensible heat flux. The number of patches depends on the length scale of the heterogeneity.

The main aim of this parameter study is to find out the response of virtual towers in heterogeneous terrain of a certain length scale with variable surface parameters, and for this reason we create two suites of simulations where each simulated case has another combination of the surface parameters. The surface parameters are the length scales $L_x$ and $L_y$ and the amplitudes $A_x$ and $A_y$. One suite is focused on kilometer scale heterogeneity, the other on hectometer scale heterogeneity. As the surface heterogeneity is two-dimensional, the length scale of the surface pattern cannot be exactly captured by a single number and therefore we concentrate on the order of magnitude of the length scale, and not on the exact length, thus comprising 4 combinations of length scales ($L_x$ and $L_y$) within the suite of kilometer scale (resp. hectometer scale) heterogeneity. For determining the average behavior under the varying surface fluxes within the suite, no weighing is applied to a particular configuration of the parameters, all amplitudes and length scales under consideration are treated equally. In Tables 2–3 we have summarized the range of the parameters that determine the landscape heterogeneity for each simulated cases within that suite (two suites of 144 simulated cases). The range of the Obukhov length and boundary-layer height expresses the variation of these quantities over the range of the parameter space spanned by the cases of the suite.

## 2.2 Control volume approach

Within the domain, we have positioned nine virtual control volumes. These control volumes are located at functionally different positions with respect to the surface heterogeneity, as can be seen in Figure 2. Four of them are located at the centers of the patches, four others are located on the borders between the patches, and one is located at the crossing of the four patches. The four at the center are positioned in a site that is homogeneous at the site scale, but heterogeneous at the landscape level. The virtual towers that are located at the borders of the patches are positioned at a site which is not homogeneous at the site level. For every control volume around a virtual tower the size is $5 \times 5$ grid points in the horizontal and 5 grid points in the vertical, representing a cube of $(50\,\mathrm{m})^3$. The limits of the control volume are set on the staggered vector grid. The implementation of the energy balance calculation for the control volumes follows the method described in Eder et al. (2015a), which incorporates the approach suggested by Wang (2010). We briefly summarize the main equation, obtained in two steps; first by spatially

averaging over the control volume, and then by additional temporally averaging over 1 hour intervals:

$$\left\langle \overline{H} \right\rangle = \left\langle \overline{w'\theta'} \right\rangle + \sum_{s=1}^{4} \left\langle \overline{\mathbf{v}'_\perp \theta'} \right\rangle_s + \left\langle \overline{w} \right\rangle \left\langle \overline{\theta} \right\rangle + \sum_{s=1}^{4} \left\langle \overline{\mathbf{v}}_\perp \right\rangle_s \left\langle \overline{\theta} \right\rangle_s + \left\langle \delta \overline{w}\, \delta \overline{\theta} \right\rangle + \sum_{s=1}^{4} \left\langle \delta \overline{\mathbf{v}}_\perp\, \delta \overline{\theta} \right\rangle_s + \left\langle \int \overline{\frac{\partial \theta}{\partial t}} \mathrm{d}z \right\rangle . \tag{5}$$

Here $H$ denotes the surface heat flux, $x$, $y$ and $z$ are the Cartesian coordinates, $w$ the wind component in $z$ direction, $\theta$ the potential temperature, $\mathbf{v}_\perp$ the velocity vector perpendicular to the lateral faces in the $xz$- or $yz$-planes, which are indicated by "$s$" during the summation over the 4 lateral faces. The angular brackets indicate the spatial average over a face of the cube, either lateral ("s"), top or ground surface, and the $\delta$ are the corresponding spatial fluctuations. An overbar indicates a temporal average and the primes the corresponding temporal fluctuations. The term on the left-hand side of the equation is the "true" surface heat flux, whereas the terms of the right-hand side denote the eddy-covariance flux at the top of the control volume, the horizontal flux divergence, the vertical and horizontal advection by the mean flow, the vertical and horizontal dispersive fluxes (Belcher et al., 2012) and the storage of $\theta$ in the control volume. The terms of the above formula are clarified in Fig. 1. A positive sign for the directional fluxes means that they point outward of the control volume. The surface flux, however, is considered positive when the flow is from the surface to the atmosphere. Where possible, the Gauß-Ostrogradski theorem[1] has been used to reformulate a divergence within the control volume as a surface term. Due to the choice of a cuboid aligned with the coordinate system for the control volume, the control volume energy balance (5) simplifies further because only the velocity component perpendicular to the faces remain. The energy balance ratio (EBR) of the control volume, which represents the amount of closure of the eddy-covariance measurement with respect to the true surface flux, is given by

$$\mathrm{EBR} = \frac{\left\langle \overline{w'\theta'} \right\rangle}{\left\langle \overline{H} \right\rangle} . \tag{7}$$

From a control volume point of view the net fluxes through the faces are what balances the storage term inside the volume, and in this manner advection effects are automatically included in the energy balance of the volume. Of course, in analogy with measurements, the fluctuations at the top face yield the "virtually measured" turbulent heat flux: first the temporal correlations are calculated, then a spatial average over the upper face of the volume is calculated. The latter average improves the statistical significance of the virtual measurement. Although the subgrid fluxes become small at the height of the control volume, we nevertheless include the vertical component of the subgrid flux into the turbulent heat flux. In this manner we can also capture the highest-frequency correlations. Real data from measurement towers is usually sampled up to $10 - 50\,\mathrm{Hz}$, whereas for computational efficiency our simulation advances with a time step of one second, i.e. our simulated data is obtained at $1\,\mathrm{Hz}$.

---

[1]The Gauß-Ostrogradski theorem or "divergence theorem" is a special case of the Stokes-Cartan theorem in differential geometry. For our purposes, we also restrict ourselves to three-dimensional space. We consider a compact volume $V$ with a piecewise smooth boundary $S$. If $\mathbf{F}$ is a continuously differentiable vector field defined on a neighborhood of $V$, then:

$$\int_V (\nabla \cdot \mathbf{F})\, \mathrm{dV} = \oint_S \mathbf{F} \cdot \mathbf{dS} . \tag{6}$$

The left side is a volume integral of the divergence of the vector field $\mathbf{F}$ over the volume V, with $\mathrm{dV}$ the volume element, and the right side is the surface integral over the boundary of the volume $V$. $\mathbf{dS}$ is the outward pointing unit normal field of the boundary $S = \partial V$ multiplied by the surface element. For our purposes we take $\mathbf{F} = \mathbf{v}\,\theta$ and $V$ is the control volume described in the text.

A higher sampling frequency would not resolve the turbulence better, as the resolution of the latter is limited by the grid spacing. The part of the total turbulent flux that is not captured directly by the resolved turbulent flux by 1-Hertz sampling is transported by the subgrid turbulent flux. For the advective components we have made a distinction between advection due to the mean flow versus advection due to the horizontal flux-divergence. In complex terrain we do not know a well-defined choice of reference for the base temperature, in contrast to the base temperature in homogeneous terrain that appeared in Webb et al. (1980). Therefore we have avoided introducing a base temperature altogether by adding up the advection by the mean flow components, this means that our advection term is the sum of the horizontal and vertical advection by the mean flow. The virtual measurement height is quite high, but this is due to the vertical resolution and the need for sufficient grid points in the vertical direction to suppress the influence of the subgrid-fluxes whence the turbulence becomes sufficiently resolved. For the integration of the temperature in the storage term we apply numerical integration with the midpoint rule, which assumes a piecewise constant interpolation function. PALM uses implicit filtering, where it is by construction assumed that the prognostic variable within the grid cell is the volumetric mean of the variable over the domain of the grid cell, therefore the midpoint rule is the most appropriate, because by definition the LES computed $\theta[k]$ is not $\theta(z = z_k)$ but instead

$$\theta[k] = \int_{z_k - dz}^{z_k + dz} \theta(z) dz \, , \tag{8}$$

with $z_k$ the height of the grid point $k$, $dz$ the grid spacing and $\theta$ the potential temperature, and we have suppressed the indices $ij$ for clarity. In this way, the summation of the LES computed discrete profile values is defined to be equal to the integration of the continuous profile,

$$\sum_{k=1}^{K} \theta[k] = \int_{0}^{z_m} \theta(z) dz \, , \tag{9}$$

with the measurement height $z_m = z_K + dz$.

## 3   Results and discussion

### 3.1   Circulation patterns in heterogeneous terrain

We start our analysis with a discussion of the location of the updrafts and downdrafts in heterogeneous terrain. For this purpose, we concentrate on a few specific cases, more precisely $A_x = A_y = 0.3$ and all four heterogeneity lengths (with $L_x = L_y$). We will take the mean vertical velocity as the simplest proxy for circulation patterns in the boundary layer. In Fig. 3 we show the time-averaged vertical velocity at the height of the control volumes (50 m). We stress that the structures at 50 m extend into the mixed layer above where the absolute velocities become larger (not shown). The reason for the additional time average (over the complete virtual measurement interval of 4 hours) of the already hourly mean data is to remove the drift of the turbulent structures. Due to the absence of a background wind, significant circulation patterns can emerge in the homogeneous case as well. With even longer averaging times a zero mean can be achieved for idealized simulations in homogeneous terrain, but in a real atmospheric boundary-layer this is not possible due to non-stationarity on those timescales.

We notice that for the heterogeneity lengths of O(km), the motions within the mixed-layer clearly reflect the surface pattern, with updrafts concentrated above the hotter patches and downdrafts above the lower patches in the 3-km heterogeneity length, and a little offset in case of the 1.5-km heterogeneity length. However, the structure of the convective turbulence for both kilometer scale are clearly different from homogeneous control run, where typical cellular convection patterns arise (Schmidt

and Schumann, 1989), though the hectometer scales are qualitatively rather similar to the homogeneous run. The latter could be a consequence of the blending height. Investigating the heterogeneity lengths of O(hm) with more horizontal detail for the time-averaged $w$, we do not see clear updrafts or downdrafts tied to the surface heterogeneity. However, in this respect it could be interesting to note that some of the hourly mean vertical velocity (without additional time-average) for the O(hm) appear better related the surface structure. Similar results appear for weaker amplitudes and also when $A_x$ is different from

$A_y$, in which case the dominant pattern is visible along the direction with the larger amplitude (not shown). We can conclude that circulations are tied to the landscape heterogeneity when it is of O(km). For the O(hm) such a correspondence is unclear. However, the latter could be related to the "coarse" grid resolution and the distance from the ground. Indeed, Mauder et al. (2010) found persistent updraft and downdraft regions during the 2008 Ottawa field campaign.

On the topic of circulations driven by a surface conditions that are by design freely convective, we investigate how the

domain average of $u_*$ is influenced by the surface heterogeneity. The ratio between the surface flux at the hottest patch and the surface flux at the coolest patch is given by:

$$r = (1 + A_x + A_y + A_x \cdot A_y) \times (1 - A_x - A_y + A_x \cdot A_y)^{-1} . \tag{10}$$

The horizontal mean of the friction velocity scales very well with the natural logarithm of this ratio,

$$u_* = -0.046 \ln(r) + 0.384 , \ R^2 = 0.85 \tag{11}$$

The remaining spread in $u_*$ does not result from the time stamp or the heterogeneity length scale. The monotonous decrease of $u_*$ in function of the heterogeneity ratio shows that for more homogeneous terrain we will obtain a slightly larger domain averaged $u_*$.

## 3.2   Virtual tower measurements for landscape heterogeneity of kilometer scale

In Fig. 4, we look at the response of the towers with respect to their location, corresponding to the simulations summarized in

Table 2. This is the average of the simulation output belonging to the suite of kilometer scale heterogeneity. In this manner, we investigate the average effect of surface heterogeneity of kilometer scale. The towers are ordered according to the available energy at their location, for our model setup the available energy is equal to the surface flux. For each tower we have plotted the energy balance residual (available energy minus the turbulent flux), the advection component from the mean flow, the flux-divergence and the storage flux, all normalized by the available energy at the respective tower, with the plot on the left collecting

the towers located in the centers of the patches and the plot on the right collecting the towers located at the borders of the patches. The normalized turbulent flux is effectively the energy balance ratio (EBR), but we show the non-closure $(1 - \text{EBR})$, i.e. the normalized energy balance residual, as the latter's magnitude is of the same size as the remaining components. The

normalized fluxes in Fig. 4 are also averaged for all the available data points of the respective tower. That is, we averaged over the data with different time stamps and also over all cases within the suite corresponding to the kilometer length scale: this entails $(6 \times 6 - 1)$ variations of the surface flux amplitudes (we do not count the case where both amplitudes are zero, $A_x = A_y = 0$, as this is a homogeneous run) multiplied by $2 \times 2$ variations of the heterogeneity length, as expressed in table 2.

The error bars on the normalized fluxes denote the spread on the virtual measurements of each tower with respect to the suite. The spread is naturally quite large since at each tower, different amplitudes for the surface heat flux pattern are considered.

To analyze the tower response in more detail, we have separated the towers at the centers (left panel) from those at the borders (central panel). We notice that most towers show the typical underestimation of the energy balance (i.e., positive energy balance residual), except for the tower located at the warmest spot where there is an updraft. In fact, the closed energy balance for the

tower in the warm patch is similar to a result in Eder et al. (2015a) where the energy balance was closed for the site with a pronounced updraft. The residual clearly depends on the location of the tower: towers located at the centers of the patches are located in a more homogeneous environment and they exhibit remarkably smaller residuals, as expected. Towers at the borders have up to 10% more imbalance than the adjacent towers in the center. The tower on the corner of the four patches has the lowest mean closure of only 69%. For towers located at the centers, it is evident that the tower sites are locally homogeneous

but there is still a clear imbalance. As a consistency check we note that the similar towers (the two towers in the center of the patches with same surface heating; the two sets of two towers on the borders between patches of similar surface heating) behave similarly. We present some arguments why the regions with updrafts have better closure. Banerjee et al. (2017) investigated the dependence of the aerodynamic resistance on the atmospheric stability for homogeneous terrain. As a consequence a surface with a higher surface heat flux is more efficient in transporting away the surface flux. Therefore, one hypothesis is that when a

patch with higher surface flux is coupled to a patch with lower surface flux in heterogeneous terrain, the patch with the higher surface flux transports part of the surface flux of the patch with the lower surface flux, due to its higher efficiency, leading to a net advection of sensible heat from the downdraft region to the updraft region. Another hypothesis is that the shape of the cellular convection cells matters: the updrafts cover a smaller area than the downdrafts. Therefore, as the turbulence structures move across the towers, above a region with preferential updrafts, the likelihood of sampling both the updrafts and downdrafts

is higher than above a region with preferential downdrafts.

In the right panel, we show the data from four homogeneous control runs (with data extraction window and data selection in the same manner as for the heterogeneous runs). Each of these simulations has nine towers as well, but now all towers have the same surface properties. The mean residual (underclosure) of the homogeneous control runs is around 10%, less than for the heterogeneous cases but not negligible. There is significant spread on the results, but the residual is mainly composed of

advection and storage. Compared to the towers at the edges (middle panel), which are locally heterogeneous, the homogeneous case is clearly different. Compared to the towers at the centers of the patches (left panel) the homogeneous case has a different average but the difference is still within the spread. It is remarkable that flux-divergence is very small in the homogeneous case, in contrast to the heterogeneous terrain. The negligible flux-divergence for a homogeneous site was also apparent in the desert site of Eder et al. (2015a).

As the residual is formed by the sum of advection by the mean flow, storage and flux-divergence, we now turn our attention to these flux components. It turns out that primarily the advection by the mean flow determines the different residuals, but that the flux-divergence has to be taken into account as well for the full picture. In addition, the storage flux also plays a role, but its signature is independent on the location of the tower, and it is always small, which is understandable for our type of surface conditions: there is only a storage flux due to the heating of the air inside the control volumes. For different towers the allocation of the residual to advection by the mean flow versus flux-divergence varies. At first the behavior of the flux-divergence appears irregular. Let us however take a closer look in Fig. 5, where the flux-divergence and advection by the mean flow, resp. are plotted against the energy balance ratio. As in Fig. 4 flux-divergence and advection are normalized by the available energy (i.e. the surface flux in our settings). In the left panel of Fig. 5 we note that the normalized flux-divergence correlates rather well to the normalized turbulent flux, when we look at their average behavior at each tower. For the individual data points the correlation is nevertheless scattered (not plotted). It is somewhat remarkable that both the towers at the center and those at the borders exhibit a similar average behavior. Indeed, the linear regression is very satisfactory when fitting the $B$-type towers and the $C$-type towers together. We could have made two separate fits, one for each tower type as in Figure 4, but with only 3 or 4 towers of different functionality a linear regression through those 3 or 4 points would carry less meaning than considering all 9 virtual towers together. If we repeat this linear regression for the advection by the mean flow versus the energy balance ratio we see that the linear correlation fits even better (Fig. 5 right panel) but that it has opposite slope. We had expected that the sum of both components correlates very well with the energy balance ratio, since the storage is small and constant, but it is an interesting result that also flux-divergence and advection separately correlate well with energy balance ratio and consequently also with each other.

Finally, we want to remark that, due to computational constraints, the virtual measurement height in our simulations lies at 50 meters, which is an order of magnitude larger than the typical tower height over short vegetation with comparable surface roughness. This means that our findings for virtual EC towers cannot be directly transferred to real eddy-covariance towers. Other LES studies of the energy balance closure point towards a larger imbalance at higher z-levels, e.g. Steinfeld et al. (2007), Huang et al. (2008), and Schalkwijk et al. (2016). It remains an open question if we can scale the measurement height (as long as it is in the constant flux layer) with the boundary-layer depth and the scale of the heterogeneity. We also analyzed the variation of EBR in function of the surface amplitudes ($A_x$ and $A_y$) but didn't find any clear dependence there.

### 3.3 Virtual tower measurements for landscape heterogeneity of hectometer scale

In Fig. 6 we repeat the foregoing analysis for the landscape heterogeneity of hectometer scale, with the parameters in the suite now corresponding to those of table 3. The difference between the towers is much less pronounced here compared to the kilometer scale. Furthermore, the towers in the center of the patches even behave in the opposite manner when the kilometer and hectometer scales are compared. Indeed, for the hectometer scales the cooler patches have a smaller residual, hence better energy balance closure, up to even a mean over-closure for the tower in the coolest patch, whereas the energy balance at the hottest patch is not closed. Another example of the opposite behavior is shown by the flux-divergence. In Fig. 6 it is positively correlated with the normalized residual, and in Fig. 7 we notice that the flux-divergence is now indeed anti-correlated with the

EBR. The advection by the mean flow is again anti-correlated with the EBR, as it was for the kilometer scale. The storage is again roughly constant for all towers. The likely cause for the different behavior between the two scales of heterogeneity would be the blending of the hectometer landscape heterogeneity due to the virtual tower heights of 50 meter. For the surface heterogeneity of $O(10^2\,\mathrm{m})$ the flux footprint of each of the towers can cover several of the surface patches, regardless of the

type of tower. Again, in the right panel of Fig. 6 we show the data from four homogeneous control runs. However, except for the flux-divergence, the tower responses in heterogeneous terrain of hectometer scale heterogeneity look similar to the homogeneous runs.

## 3.4 Correlations with the energy balance ratio

We investigate the possible connection between the energy balance ratio, the different flux contributions, and variables such

as friction velocity and boundary-layer height. We performed a linear correlation analysis between these variables and the energy balance ratio. We made one restriction on the data set, which is to limit the boundary-layer depth to values larger than 1 km, thereby excluding about 8% of the data, in order to obtain a better representation of the boundary-layer depth (when the boundary-layer depths smaller than 1 km are included, the correlation deteriorates).

We found that friction velocity and boundary-layer depth cluster are well-correlated with each other, but not with EBR. Al-

though we might have supposed that higher boundary-layer heights will arise if patches are present with vigorous surface heating, however we found that $u_*$ decreased with stronger surface heterogeneity. Closer analysis reveals that the highest boundary layer heights are obtained when the heterogeneity amplitudes are smaller and the domain is more homogeneous. Hence the former clustering can be explained because in our scenario with varying heterogeneity amplitudes the highest boundary-layer height and larger $u_*$ are both obtained for smaller heterogeneity amplitudes. Though advection and flux-divergence correlate

well with EBR, they cannot be measured independently and therefore cannot be used as independent predictors. In the literature (e.g. Stoy et al., 2013; Eder et al., 2015b) a correlation between friction velocity and energy balance closure has been found: a high friction velocity leads to a smaller residual. Typically, a higher friction velocity is correlated to smaller atmospheric instability, and hence roll-like convection instead of cellular convection. Maronga and Raasch (2013) found that boundary-layer rolls "smear out" the surface heterogeneity, leading to an effective surface that looks less heterogeneous, which has been related

to a higher EBR (Mauder et al., 2007; Stoy et al., 2013). Therefore a possible cause for the present low correlation of $u_*$ with the EBR, could be our range of the stability parameter. For the free convective cases considered here the stability parameter lies below the range where the friction velocity has a high correlation with EBR.

The linear correlation analysis shows that the simulated EBR does not linearly depend on easily measured characteristics. As we have learned from Fig. 5, there can be a good fit between the parameter-averages of two variables, e.g. normalized

flux-divergence and energy balance ratio average, despite the fact that the individual data points do not correlate as well. This highlights the importance of testing parameterizations for the energy balance closure problem on the level of a data ensemble, instead of parameterizing on the level of the individual hourly measurements.

## 4 Conclusions

In this work, we have investigated the effect of idealized surface heterogeneity on the components of the surface energy budget measured at virtual measurement towers, by means of large-eddy simulation. By means of a control volume approach, we have decomposed the modeled surface energy budget to highlight its partitioning, and we have shown that the modeled energy balance ratio exhibits values that are found in field experiments. In addition, this approach allows us to investigate the energy balance closure in two-dimensional complex terrain. We have found that for surface heterogeneity with length scale of order kilometer, there is a clear relation between the energy budget components and the location of the tower with respect to the patches of surface heterogeneity. For surface heterogeneity of hectometer scale, the response of the different towers appears to depend to a lesser extent on their respective location. Towers located at the borders between patches with different surface heat flux have worse closure than towers located in the center of a patch. Although storage terms are not negligible, the size of the residual depends mostly on the advection and flux-divergence terms. Remarkably, flux-divergence and advection by the mean flow correlate separately very well with the energy balance ratio, which implies that the EBR can be explained by the advection or flux-divergence only, as the latter two are well correlated among themselves. For the kilometer scale heterogeneities, advection by the mean flow and flux-divergence behave opposite, while they are positively correlated for hectometer scale heterogeneities. We did not find a high correlation between the friction velocity and energy balance ratio, but this could be due to the limited range of $u_*$ as we have investigated free convection.

## Appendix A:  Example of the heterogeneity length scale of a field site

Even though the focus of this study is on virtual flux measurements, we can look at an example of a real EC measurement site to make a qualitative comparison of these virtual tower measurements with real tower measurements. In a first approximation, the heterogeneity of the landscape around a measurement site can be characterized by the dominant length scale of a suitable surface variable. In Eder et al. (2014), the dominant length scales corresponding to a few sites belonging to the TERENO measurement network (Zacharias et al., 2011) were computed from the Fourier spectrum of the surface roughness. The site of them with the least pronounced topography, the site Fendt, has an effective length scale close to 3 km and a mean EBR of 0.77, which is a typical value for the energy balance ratio (Stoy et al., 2013). The location of the measurement tower in Fendt would correspond to a tower of the central type, and due to its location in the meadow with lower albedo than the forest or the small built-up area, we would assign it to the central tower of the cooler patch. However, the Fourier spectrum of the sensible heat flux may differ from that of the surface roughness. Moreover, the Fourier spectrum for the TERENO site in Fendt exhibits an additional local maximum in its Fourier spectrum of the surface roughness, at 600 m (Eder, pers. comm.). Additionally, it should be noted that even a simplified version of the landscape heterogeneity of Fendt would appear primarily striplike, in contrast the synthetic chessboard pattern here. The EC tower of Fendt is located in a large north-south oriented meadow which is flanked by two forests further away to the west and the east. Despite these apparent differences between our idealised simulations and the real situation at the Fendt site, the EBR of 0.77 is comparable to the EBR of the virtual towers investigated here for the kilometer heterogeneity.

*Competing interests.* There are no competing interests.

*Acknowledgements.* This work was conducted within the Helmholtz Young Investigators Group "Capturing all relevant scales of biosphere-atmosphere exchange — the enigmatic energy balance closure problem", which is funded by the Helmholtz-Association through the President's Initiative and Networking Fund, and by KIT. We thank the PALM group at Leibniz University Hannover for their open-source PALM code and their support.

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

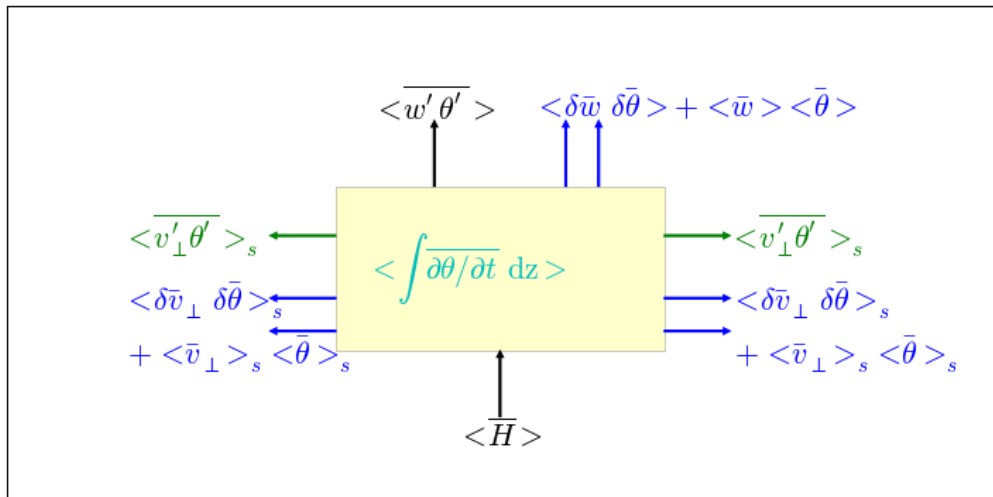

**Figure 1.** Graphical representation of (5). The control volume is colored in yellow, with horizontal flux-divergence in green, the advection terms in blue, and the storage flux in cyan. The surface flux and the measured turbulent flux are both in black. For clarity the lateral dimension perpendicular to the cross-section is not shown. The direction of the arrows indicate a positive contribution.

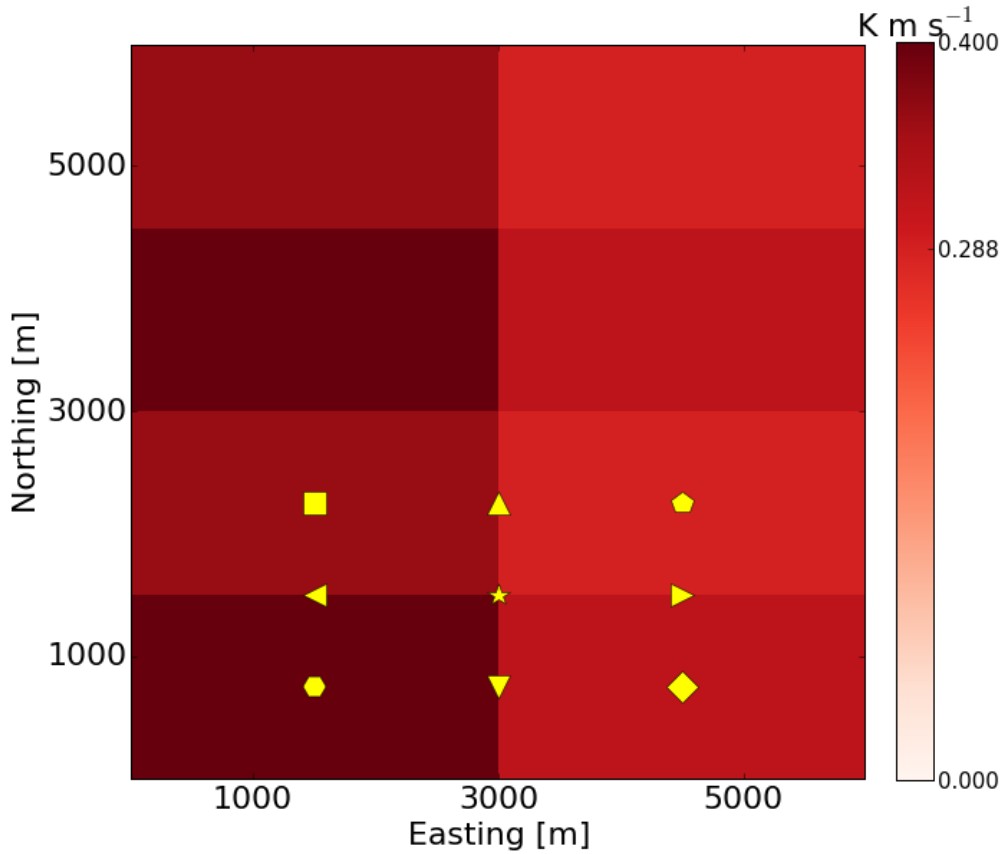

**Figure 2.** Fixed location of the virtual towers for the kilometer scale heterogeneity. The surface heat flux pattern of this example corresponds to $L_x = 3000\,\text{m}$, $L_y = 1500\,\text{m}$. Please note that all the control volumes have the same shape of $5 \times 5 \times 5$ grid points, the symbols are only to distinguish the different types of towers. For the hectometer scale heterogeneity, the towers are located at the similar positions in or in between the patches, only the patches are smaller. The towers fall into two classes: those located at the center of the patches and those located at the borders.

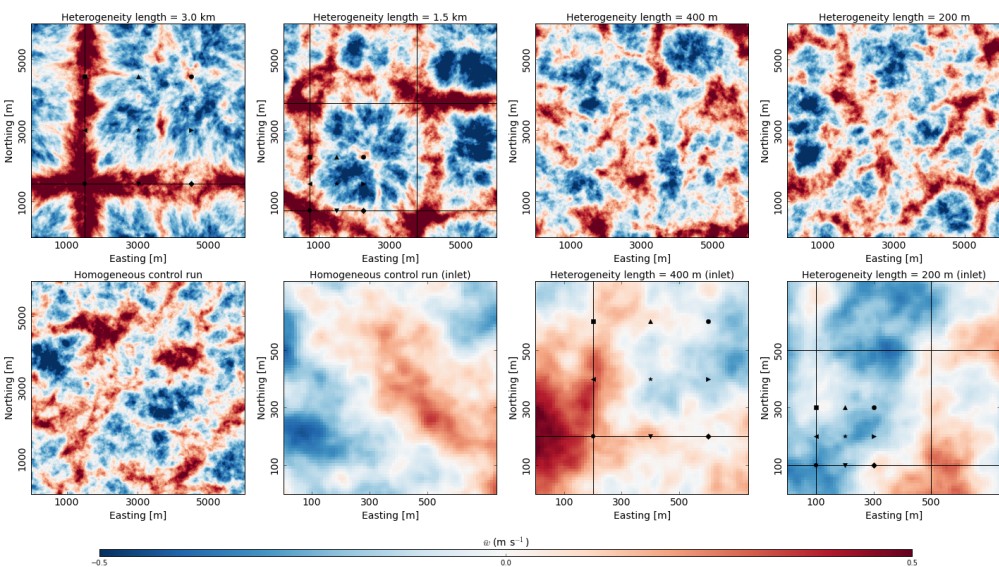

**Figure 3.** Analysis of the circulation patterns induced but the surface heterogeneity by means of the vertical velocity ($w$) averaged over the 4-hr data output, including a homogeneous control run. The results are for a particular surface amplitude of $A_x = A_y = 0.3$ and with $L_x = L_y$ ($z = 50$ m). For reference the tower locations are indicated as well, as is the center of the "hot" patches by means of a black line. The plots of the whole domain for O(hm) show their reminiscence with the homogeneous control run. For the O(hm) heterogeneity we show an inlet around the towers, because the correspondence with the surface heterogeneity is otherwise hard to visualize due to the smallness of the heterogeneity length.

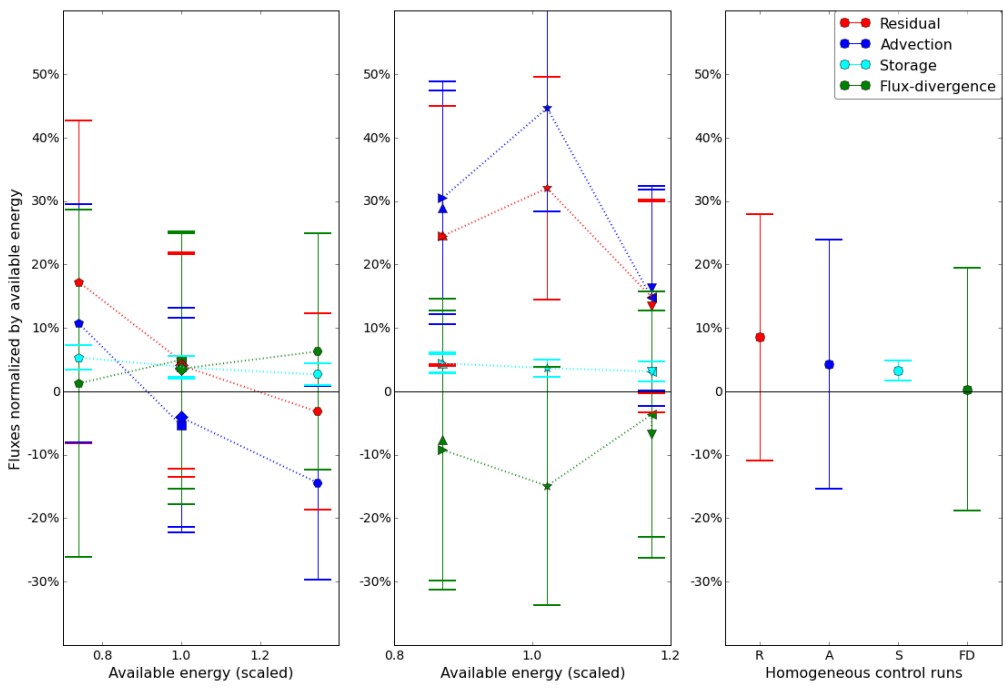

**Figure 4.** Control volume fluxes as a function of available energy (scaled by the median value) for kilometer scale landscape heterogeneity. The fluxes are normalized by the available energy at their respective location, in our setup this means normalization by the surface flux. Please note that we have plotted the non-closure (normalized energy balance residual) instead of the energy balance ratio EBR (normalized turbulent flux). The left panel shows the towers at the centers of the patches, the middle panel the towers at the edges of the patches, and the right panel the results for the homogeneous control runs. For the tower symbols, see Fig. 1. The error bars denote the spread over the different cases of surface heterogeneity within the suite of kilometer scale surface heterogeneity. The abscissa is the available energy at the tower, but scaled by the mean available energy of the nine towers for that case. In this way, we can group the towers by tower type, also for the cases with different surface amplitudes. Thus, the low values represent the towers located at the cooler patches (downdrafts), the high values the towers located at the hotter patches (updrafts). See text for further discussion.

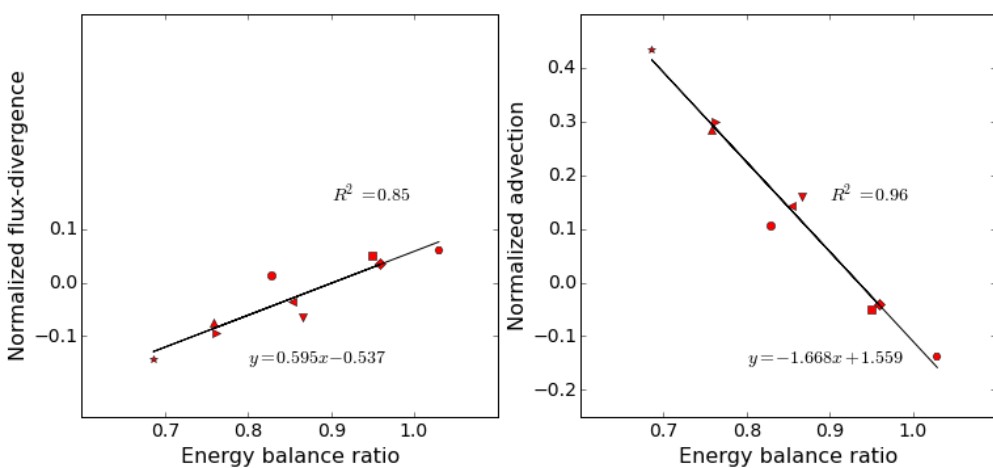

**Figure 5.** Correlation between flux-divergence and EBR for kilometer scale heterogeneity (left panel); correlation between advection and EBR for kilometer scale heterogeneity (right panel)

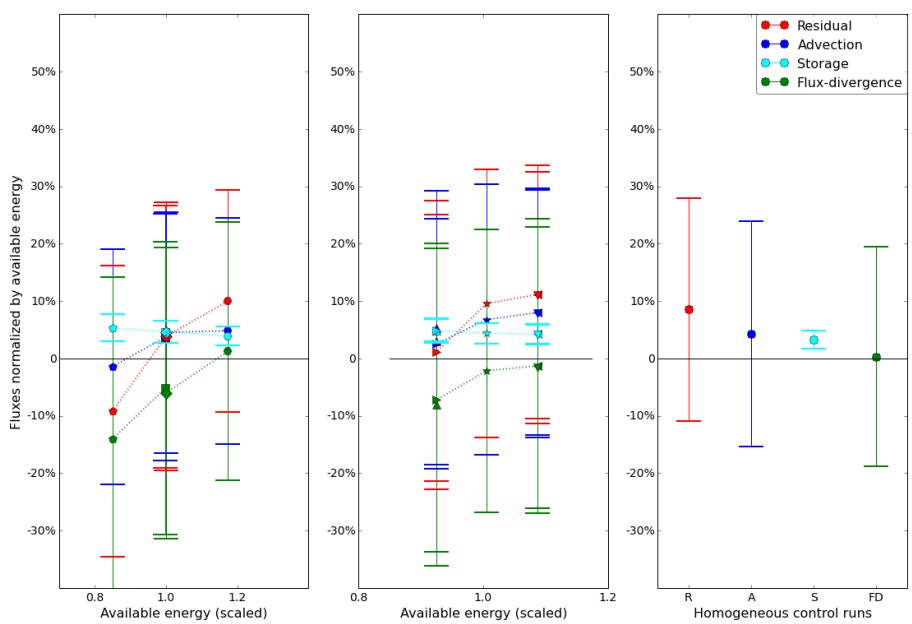

**Figure 6.** Control volume fluxes as a function of available energy (scaled by the median value) for hectometer scale landscape heterogeneity. The fluxes are normalized by the available energy at their respective location, in our setup this means normalization by the surface flux. Please note we have plotted the non-closure (normalized energy balance residual) instead of the energy balance ratio EBR (normalized turbulent flux). The left panel shows the towers at the centers of the patches, the middle panel the towers at the edges of the patches, and the right panel the results for the homogeneous control runs. For the tower symbols, see Fig. 1. The error bars denote the spread over the different cases of surface heterogeneity within the suite of hectometer scale surface heterogeneity. The abscissa is the available energy at the tower, but scaled by the mean available energy of the nine towers for that case. In this way, we can group the towers by tower type, also for the cases with different surface amplitudes. Thus, the low values represent the towers located at the cooler patches (downdrafts), the high values the towers located at the hotter patches (updrafts). See text for further discussion.

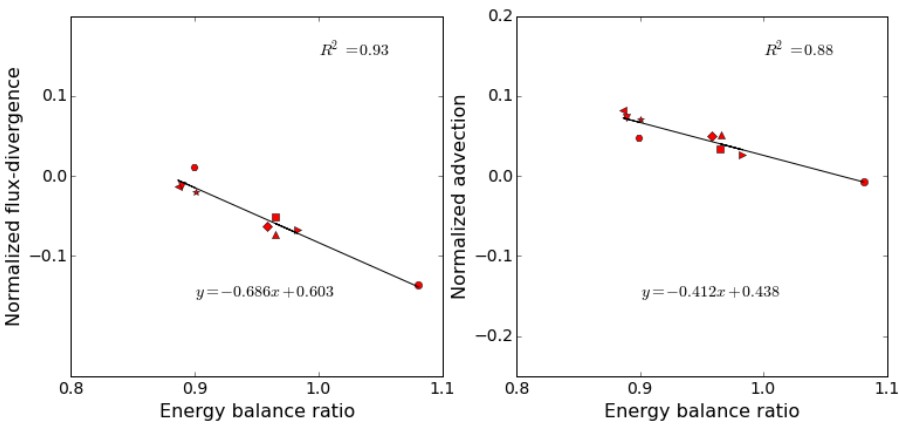

**Figure 7.** Correlation between flux-divergence and EBR for hectometer scale heterogeneity (left panel); correlation between advection and EBR for hectometer scale heterogeneity (right panel)

**Table 1.** Parameters of the LES configuration

| Quantity | Unit | Value |
| --- | --- | --- |
| Number of grid points | (-) | 600 x 600 x 240 |
| Spatial resolution (dx, dy, dz) | (m) | 10.0, 10.0, 10.0 |
| Domain size | (m$^3$) | 6,000 x 6,000 x 2,400 |
| Temporal resolution | (s) | 1.0 |
| Spin-up time | (s) | 7,200.0 |
| Data capture | (s) | 7,200.0 - 21,600.0 |
| Averaging interval | (s) | 3,600.0 |
| Size of the control volume | (m$^3$) | 50.0 x 50.0 x 50.0 |
| Approximate walltime for one simulation | (core-hours) | 5500 |
| Roughness length | (m) | 0.1 |
| Surface moisture flux | (kg kg$^{-1}$ m s$^{-1}$) | 5e$^{-6}$ |

**Table 2.** Parameters of the simulations within the suite focusing on the landscape heterogeneity at kilometer scale

| Strong to free convection | | $6 \times 6 \times 2 \times 2 = 144$ cases |
| --- | --- | --- |
| Average surface heat flux $(H_0)$ | $(\text{K m s}^{-1})$ | 0.25 |
| Amplitude $x$ $(A_x)$ | $(\text{K m s}^{-1})$ | 0.0 ; 0.1 ; 0.2 ; 0.3 ; 0.4 ; 0.5 |
| Amplitude $y$ $(A_y)$ | $(\text{K m s}^{-1})$ | 0.0 ; 0.1 ; 0.2 ; 0.3 ; 0.4 ; 0.5 |
| Length scale $x$ $(L_x)$ | (m) | 1,500.0 ; 3,000.0 |
| Length scale $y$ $(L_y)$ | (m) | 1,500.0 ; 3,000.0 |
| Surface flux range | $(\text{K m s}^{-1})$ | From 0.0625 to 0.5625 |
| $u_*$ (momentum flux) | $(\text{m s}^{-1})$ | From 0.071 to 0.69 |
| Boundary-layer height | (km) | From 1.4 to 2.2 |
| Obukhov length | (m) | From -36.1 to -0.04 (average -3.96) |

**Table 3.** Parameters of the simulations within the suite focusing on the landscape heterogeneity at hectometer scale

| Strong to free convection | | $6 \times 6 \times 2 \times 2 = 144$ cases |
|---|---|---|
| Average surface heat flux ($H_0$) | (K m s$^{-1}$) | 0.25 |
| Amplitude $x$ ($A_x$) | (K m s$^{-1}$) | 0.0 ; 0.1 ; 0.2 ; 0.3 ; 0.4 ; 0.5 |
| Amplitude $y$ ($A_y$) | (K m s$^{-1}$) | 0.0 ; 0.1 ; 0.2 ; 0.3 ; 0.4 ; 0.5 |
| Length scale $x$ ($L_x$) | (m) | 200.0 ; 400.0 |
| Length scale $y$ ($L_y$) | (m) | 200.0 ; 400.0 |
| Surface flux range | (K m s$^{-1}$) | From 0.0625 to 0.5625 |
| $u_*$ (momentum flux) | (m s$^{-1}$) | From 0.052 to 0.74 |
| Boundary-layer height | (km) | From 1.5 to 2.2 |
| Obukhov length | (m) | From -53.8 to -0.01 (average -5.75) |