# Peer review of "The influence of idealized surface heterogeneity on virtual turbulent flux measurements"

_Atmospheric Chemistry and Physics, 2017_

## Referee Comment (RC1) · Anonymous Referee #1 · 8 Oct 2017

Review of
**The influence of idealized surface heterogeneity on virtual turbulent flux measurements**

by
Frederik De Roo and Matthias Mauder

October 7, 2017

**General comments:**

In field experiments, it is typically assumed horizontally homogeneous and steady-state conditions, and the vertical turbulent flux of heat is measured as an estimation of the surface heat flux. However, when computing the total energy budget (including radiation and latent heat), the vertical turbulent flux of heat does not close the energy budget, indicating that one or both assumptions are wrong. In this study, it is investigated (i) the effect of surface heat flux heterogeneity on the difference between the vertical turbulent heat flux and the surface heat flux itself (the residual of the energy budget in field experiments), and (ii) the correlation between simulation variables that can be obtained in field experiments and this residual. It was found that in the presence of horizontal heterogeneity in the sensible heat flux in the order of hectometer and kilometer, there is indeed a difference (a residual) between the surface flux and the vertical turbulent flux at 50 m. Estimations located at the edge of patches with different surface heat fluxes have higher residual than the ones located at the center of patches. The effect of storage was smaller than the effects of advection and flux-divergence. A correlation between the residual and the difference in temperature between surface and turbulent flux measurement height was observed.

This work focuses on a very interesting and useful topic, and it uses an appropriate tool to investigate it (LES). However, the simulation itself is not properly discussed and the results obtained are not extrapolated to the real case, therefore contributing very little to the advancement of the knowledge in the field. In addition, the estimation of correlations between the residual and the atmospheric parameters needs to be improved. For this reason, I would like to suggest major revision including (i) a thorough discussion of the simulation results, (ii) a correction of the correlation estimations, and (iii) a proposition of a model that could be used in field experiments (or some other useful information for future field experiments), in order to improve the impact of the manuscript.

Because this is my first time thinking about this topic, I may not be able to give the best suggestions about how to present more useful results, although I will try my best. Nevertheless, I do know that there is a rich

amount of information given by the LES, so it should be possible to extract a lot of interesting information from it. It would be interesting to try to obtain results that would help people from field experiments to either understand what is likely causing the closure problem, to better plan for future experiments, or to improve the closure problem.

**Specific comments:**

1. The LES model: despite being a well-known technique and a largely used model, I think a little more information about the PALM-LES should be given. This is a simulation of free convection, in which no mean streamwise velocity is present. Have this type of simulation been performed with PALM-LES before? If so, a citation and a brief summary of model's performance should be given. Otherwise, some description of the velocity and temperature fields should be given, including an assessment of the level of reality being represented by the model.

2. The simulations: all the information needed to reproduce the simulations exactly should be given. For example, the exact values of initial and boundary conditions of all variables, the strength of the inversion and the subsidence, etc. I'm still confused about how many simulations were run. I'm assuming it was two, one for the kilometer and another for the hectometer case. If so, the information in Tables 2 and 3 are confusing. Does the word "cases" mean "patches"? What are the ranges in ABL height and Obukhov length, are they in time or space? Are these ranges resonable?

3. p. 6, l. 21–22: "the Gauß-Ostrogradski theorem has been used to reformulate a divergence within the control volume as a surface term", please be explicit on what was done.

4. The PCA analysis: in the Methods section, a brief description of the PCA method and how to interpret its results should be given. Right now this is completely left to references, but I think I should be able to understand the technique and the plot overall without having to look in another paper. Also, a more exact description of what was done should be given, including how many and which variables were used, which equation or software, etc.

5. Simulation results: the simulation results should be presented and discussed before presenting the statistics. For example, how do the spatial fields of temperature and heat flux look like, and where do the towers rest in this field? Some time series at the tower place, to see what the towers are measuring and what are the scales of motion in it. How do they compare with the scales of heterogeneity? How much of the fluxes are resolved compared to the sub-grid scale? How realistic are these simulations? A thorough discussion of the simulation is definitely needed, as it would help to discuss the physics of the results presented later.

6. Discussion: there are three distinct physical phenomenon that could be impacting the residual: transport of the mean field, transport of the fluctuating field, and storage. What are the physics involved in

each process, how is the simulation capturing them, and how do they look like in the simulation? Is it realistic to look into the advection effect, for example, in a simulation that has no mean streamwise advection? What happened to the vertical and horizontal dispersive fluxes?

7. p. 7, l. 19: what is the *available energy*? It is the reference value in the results section, but there is no definition of it. It is only explained in the Fig. 2 caption, but it should be clear in the text too.

8. p. 7, l. 19: please make it clear if the *advection* term includes both horizontal and vertical advection.

9. p. 7, l. 29–31: "We notice that most towers show the typical underestimation of the energy balance, except for the tower located at the warmest spot where there is an updraft." How do you know it is an updraft? Is it always an updraft? On average? It is mentioned in the abstract that updraft and downdraft positions were chosen for the towers, but this is not showed or discussed in the paper. Do constant regions of updraft/downdraft exist in the simulation or in reality?

10. p. 7, l. 30–31: please explain better physically the causes of a negative/positive residual, and why there is a negative residual where there is an updraft.

11. p. 8, l. 13–14: "In the left panel of Fig. 4 we note that the normalized flux-divergence correlates rather well to the normalized turbulent flux, when we look at their average behavior at each tower." What does "normalized" mean? If it is the "normalized by the available energy at their respective location" mentioned in the caption of Fig. 2, it should be mentioned in the text too.

12. p. 8, l. 22–23: "flux-divergence and advection separately correlate well with energy balance ratio and consequently also with each other". What does that mean physically? Does it makes sense that these two processes are correlated in the simulation? What is the implication of this for the imbalance observed in the real case?

13. p. 8, l. 24–26: "Finally, we want to remark that, due to computational constraints, the virtual measurement height in our simulations lies at 50 meters, which is an order of magnitude larger than the typical tower height over short vegetation with comparable surface roughness. This means that our findings for virtual EC towers cannot be directly transferred to real eddy-covariance towers." Why not compare the residual observed in the simulation as a function of height, including the lower points? If I understood correctly, the residual term is estimated at a given height, so it does not need the control volume approach. If so, you can see if the results at $50\,\mathrm{m}$ are similar at $10\,\mathrm{m}$, and if the conclusions could be extrapolated. Without this extrapolation, I see little usefulness in the conclusions obtained here.

14. p. 8, l. 32–33: "the towers in the center of the patches even behave in the opposite manner when the kilometer and hectometer scales are compared". This needs to be better investigated and discussed physically. What can be causing this?

15. Discussion: I think it is important to discuss the differences when there is a small residual due to low values of all other fluxes, or due to their canceling effect. What is likely to be happening in field experiments?

16. p. 9, l. 4–7: "The likely cause for the different behavior between the two scales of heterogeneity would be the blending of the hectometer landscape heterogeneity due to the virtual tower heights of 50 meter. For the surface heterogeneity of $O(10^2$ m) the flux footprint of each of the towers can cover several of the surface patches, regardless of the type of tower." I agree this is likely the cause of difference, but how does this make the relation between flux-divergence and EBR opposite, or the relation between the residual and the surface flux opposite?

17. PCA results: from the very little I know about PCA analysis, I think the results obtained here are not useful. I believe it is useful to look at correlation biplots when most of the variance are explained by the first two PC's (something around 90%), not the 60% found here. As explained by Greenacre (2010) (your own reference on the topic), PCA correlation biplots are useful when there is a clear separation in the scree plot between the first two PC's and the rest (called *elbow*), which is definitely not the case here. In your data, the third PC is almost as important as the second, and it is not taken into account. In addition, the EBR (the variable you want to explain) is the one with less representation by the first two PC's among the variables in the correlation biplot, being much less than 50% representation in the hectometer case. I don't know which analysis should be done instead, but I think PCA is not the one.

18. Conclusions: ignoring the results from the PCA, the only conclusion from this work is that this type of heterogeneity in the surface flux generates a difference between the turbulent flux measured at 50 m and the surface flux itself. This is an interesting information, but given the potential of these simulations, I believe that much more can be obtained. I would like to see some results and conclusions that could help improve or understand the closure problem in field experiments. The idea of finding parameters measurable in the field that correlate well with the residual is a good approach. But another technique should be used to find the right parameters. After finding them, I suggest that a model for the residual should be developed and tested against the LES results. Naturally, in reality things are not equal to the LES, but this would gives us a place to start. When combined with a thorough discussion of the simulation itself, it would be easier to extrapolate the results and conclusions to the real case. Without it, I don't see any significant contribution to the field.

**Technical corrections:**

- Abstract: there is too much introduction information in this abstract. It is also clear from it that there is no significant contribution from this work. After improving the results and conclusion, the abstract should focus more on them.

- p. 4, l. 17: "we have added a very small moisture flux" why? why not make it zero?

- Figure 1: missing unit of color plot (surface heat flux)

- Equation (3): a sketch of the fluxes and a figure showing where in the 50 by 50 m box each term is being calculated would be usedful.

- p. 7, l. 30: include text: "most towers show the typical underestimation of the energy balance **(i.e., positive energy balance residual**)"

- p. 8, l. 12: what is "resp."?

- p. 8, l. 12, 13: it should be Fig. 3 instead of Fig. 4

- p. 8, l. 20: opposite slope? They look the same to me...

- p. 8, l. 31: since you are comparing Fig. 2 with Fig. 4, why not make them one single plot? It makes it easier to compare.

- Fig. 3 and Fig. 5 are equal, one of them is wrong (I guess it is Fig. 3, based on the text)

- p. 9, l. 2: Fig. 4 should be Fig. 3

**References**

Greenacre, M. (2010). *Biplots in Practice*. fundacion BVBA.

---

## Referee Comment (RC2) · Anonymous Referee #2 · 13 Nov 2017

Manuscript number: acp-2017-498
Title: **The influence of idealized surface heterogeneity on virtual turbulent flux measurements**
Authors: F De Roo and M Mauder

Reviewer recommendation: *major revisions.*

**Summary:**

The authors investigate a long-standing problem: the non-closure of the energy balance observed in eddy-covariance measurements. The employ large-eddy simulations over heterogeneous terrain to perform measurements of the energy balance for virtual towers located in their domain, while varying the surface heterogeneity from case to case. In general I believe their results might be interesting to the research community, but I do have some critical comments that need to be addressed first.

First, their analysis is based on an energy balance formulation which is (atleast from dimensional point of view) not correct. Second, their results seem to indicate that advection due to surface heterogeneity plays a key role for the imbalance, but they do not compare their results against a homogeneous control run to prove that their results are not an artefact of insufficient averaging. Third, their analysis is limited to a height of 50 m above ground, which is way higher than the height of eddy covariance measurements. Fourth, the length of the paper (the number of figures) appears too long for the rather short message of it. To me, this research is better suited to be published as a research note (mainly because of my fourth point). And lastly, there are language issues, including wrong use of English language,ambiguous statements, and formatting issues. It makes the reading of the manuscript a bit hard so the authors should make some effort to formulate precise and correct sentences.

I hence recommend to reconsider this paper for publication after major revisions.

A list of more detailed comments are given below.

**Detailed comments by the reviewer:**

Major comments:

1. Energy balance (Eq. 3)
   I am struggling with this equation due to several reasons. First, the dimensions do not fit. A flux should be given in K m/s. The storge term, however, is the integral of $d\theta/dt$ over all three dimensions in space so that the resulting dimension is K / s * m³ = K m/s * m², which is NOT a flux. The equation thus can't be correct (this applies to the original equation in Eder et al. 2015 as well). Second, the authors refer to the second right hand side term as the flux divergence. That is not correct. The divergence involves the derivation in space, but here only a sum (difference) is calculated. Using the divergence, the equation would start to make sense, as the dimension of a flux divergence is K/s, which fits to a storage term ($d\theta/dt$). Third, to me it would make more sense to put storage on the left hand side and the rest on the right hand side. Fourth, why is the storage term integrated over a volume? Which volume? Shouldn't ALL terms be integrated over the entire control volumes? Is it really an integral or an averaging? Moreover, you name the $\langle w \rangle \langle \theta \rangle$ terms "vertical advection" and so on. But a turbulent flux is also advective, so this is not correct language. You mean the advection by the mean flow, which the flux divergence represents the advection by the turbulent flow?! Lastly, The summation over the lateral fluxes seems to work, but I expect the same summation over the vertical fluxes, i.e., $w'\theta' + H$, but even in the equation it seems to be a difference and not a sum ($w'\theta' - H$ if you put all terms on one side). In Summary, the equation is very difficult to understand, and reading the Eder et al. paper does not really contribute to the understanding. Finally, I am also not sure that you can apply your averaging procedure for such a large control volume. Temperature, e.g. has a logarithmic profile near the surface, so you must

be careful when integrating the storage term over all control volumes.

2. Homogeneous control run
   From Fig. 2 it becomes obvious that the EC flux cannot make up for the available energy. I think that is the main point of the paper. It is shown nicely, that advection (of the mean flow) plays an important role, and also flux divergence. You relate this to the heterogeneous surface, but you do not prove it. What you need to do is to add results from control runs with a homogeneous surface. What do you expect to happen? At least advection should go to zero, because all velocity components should go to zero when sufficiently averaged.

3. Measurement height
   I do understand why you did your virtual measurements at an elevated height because of a relatively coarse grid spacing and you need 5 grid volumens. However, I could not find the grid spacing in the text, so I had to infer it from the table.The main point, however, is – and you state that yourself – that EC measurements are performed much closer to the surface (2 m) so that it is completely unknown whether your results are transferable to 2 m height. Maronga and Raasch (2003) showed, e.g., that the effect of secondary circulations (heterogeneity effect) tends to zero close to the surface and is most pronounced at upper levels. To me it remains unclear what the benefit of your study is to the research community.

4. Figures and length
   I suggest to shorten the paper and stick to the main message. Not all figures are needed I think. Especially the "special plots" are difficult to read given the rather simple message you want to convey. By the way, figure 1 could be plotted nicer.

5. Language
   You should be more careful with language editing before submission. I understand that correct English is an issue for non-native speakers, but please avoid wrong syntax and incorrect formatting. Examples: P1 L4: there is an extra "was", at several points you use the phrase "in function of", which to my knowledge simply does not exist, P7 L14: "Table table:2", wrong formatting. Referees spend their rare time for reading your manuscript without having any benefit from it. Beeing confronted with such carelessness suggests that the authors did not read their own manuscript carefully – and this does not really motivate to review your manuscript in-depth. See my detailed language corrections below.

Minor comments:

1. P2 L17: Why is there a correlation between the friction velocity and the energy balance closure? Also: there is nothing as a "good" correlation. It has a value, and can be possible regarded as "high".

2. P2 L19: You talk about secondary circulationy by self-organization, but I am not sure this contradicts the definition of a secondary circulation. Imagine a self-organized flow (give an example!), then what is the difference between primary and secondary circulation? I consider the hexagonal patterns for instance as the typical case of self-organization in the CBL, but that is what we call the primary circulation.

3. P2 L29: Secondary circulations do not "decay" to zero at the surface! They are simply not there.

4. P3 L5: Why is w=0 at first grid level?

5. P3 L19: The statement does not make sense. You can't say that a different partitioning of sensible and latent heat fluxes led to a higher Bown ratio, because that is simply the definition of the Bowen ratio!

6. P4 L19: What is "strong convection"? All your simulated cases are without mean wind, so you have free convective conditions in all your simulations.

7. P5 L16-31: You state that Neumann conditions are used at the lower boundary, which means you are prescribing surface fluxes. Then, how to you prescribe the surface momentum fluxes? With what values? Somehow you need to take into account the surface roughness, but for that you will need to use Monin-Obukhov Similarity Theory.

8. P6 L4: How can a tower be homogeneous? The surface around its base can, though.

9. P7 L14: There is no dataset in the table.

10. P7 L31: "worst imbalance of only 69%". I guess you mean closure, not imbalance.

11. P8 L1: finally you mention that you are talking about the "mean advection", but it should read "advection by the mean flow".

12. P8 L25: You start the second sentence nearly exactly as you have finished the preceding one. That sounds odd.

13. P8: You discuss the difference between kilometer and hectometer scale heterogeneity; but from previous studies it is known that only those can trigger secondary circulations whose scale is in the order of magnitude of the boundary layer depth. Now I am wondering whether you took this into account or not. In the end, you do not show secondary circulations at all, so it remains a secret whether your findings are related to the scale of the heterogeneity of the ratio of the scale of the heterogeneity to the boundary layer depth. Or do you see local circulations that were not seen in previous studies? If yes, you should show them. You only provide very little evidence here.

14. P9 L13: Strictly, the friction velocity is zero in free convection as the mean wind is zero, so how can it increase for stronger circulation patterns? You must describe that you treat u* as a local quantity and that the primary circulation creates local wind shear near the surface (if that is what you mean).

Language / typos (just an outline):

1. P1 L1: "pending problem" - language.
2. P1 L4: extra "was", "boundary-layer scale"
3. P1 L14: "order of magnitude" instead of "decade"
4. P2 L1: "Earth's"
5. P5 L3: influence on what?
6. P5 L18: (and throughout the text): "in function of" -> "as a function of"?
7. P6 L6: what is the "?" for?
8. P6 L27,29: Decide: "Hz" or "Hertz"
9. P7 L19: "time steps"
10. P7 L24: "we now plot" - rewrite.
11. Fig. 2: what does "in casu" mean?
12. Tab. 2: What does "1.4 ? 2.2" mean?
13. Tab. 2: "-36,1" should read "36.1",
14. Tab. 2: I'd more like a long-list with all simulations
15. Tab. 2+3: What is the Boundary-layer height in "short" (1.4 – 2.2), while the Obukhov length is in "long" (From ... to ...)?

---

## Author Comment (AC1) · 30 Dec 2017

**Reply to the reviewers**

Frederik De Roo and Matthias Mauder

December 2017

*We thank the reviewers for their careful reading and constructive comments. We think that our implementation of the comments has improved the discussion paper and made it clearer for a wider audience.*

*Our responses are in italic, and our changes in the manuscript are straight. We have used a brown color for the part of the comments of reviewer 2 that we already incorporated in the original Discussion paper, and a blue response for the recent changes.*

**1  Reviewer 1**

**1.1  Decision**

For this reason, I would like to suggest major revision including (i) a thorough discussion of the simulation results, (ii) a correction of the correlation estimations, and (iii) a proposition of a model that could be used in field experiments (or some other useful information for future field experiments), in order to improve the impact of the manuscript.

*We improved our description of the model and added material to the discussion of the simulation results and correlation analysis.*

**1.2  Specific comments**

1. The LES model: despite being a well-known technique and a largely used model, I think a little more information about the PALM-LES should be given. This is a simulation of free convection, in which no mean streamwise velocity is present. Have this type of simulation been performed with PALMLES before? If so, a citation and a brief summary of models performance should be given. Otherwise, some description of the velocity and temperature fields should be given, including an assessment of the level of reality being represented by the model.

   *We expanded the section on PALM, its simulation approach, and the boundary conditions and initial settings.*

   **Reworked paragraphs in the manuscript (section 2.1):**

For our simulations we have made use of the LES model PALM (Maronga et al 2015). More precisely, we ran our simulations with PALM version 3.9. PALM resolves the turbulence down to the scale of the grid spacing, all turbulence below is parameterized by implicit filtering. The closure model in PALM is a so-called 1.5-order closure scheme, where the equations for the resolved velocities and scalars are derived by implicit filtering over each grid box of the turbulent Navier-Stokes equations, and where an additional prognostic equation for the turbulent kinetic energy is solved. The turbulent kinetic energy in PALM (the sum of the variance of the subgrid-scale velocities) allows to model the energetic content of the subgrid-scale motions, and because it is related to spatial filtering it should not be confused with the typical turbulent kinetic energy in eddy-covariance measurements related to the averaging of a time series. Of course, the latter can be approximated by the resolved kinetic energy in PALM plus the subgrid-scale turbulent kinetic energy. Finally, the Reynolds fluxes that appear in PALM's filtered equations (the spatial covariances of the subgrid-scale quantities) are parameterized by a flux-gradient approach involving the resolved gradient and a diffusivity coefficient that depends on the before-mentioned turbulent kinetic energy, the grid spacing and the height above the lower surface. However, at the first gridpoint above the surface, Monin-Obukhov similarity theory is applied and therefore the turbulence there is completely parameterized.

Relevant parameters of the simulation setup are summarized in Table 1, the grid spacing is 10 m in all three dimensions and the domain size is $6 \times 6$ square kilometers in the horizontal, and 2.4 km in the vertical. The boundary conditions of the simulations are periodic in the lateral dimensions. For the velocity we have Dirichlet conditions at the bottom (i.e. rigid no-slip conditions) with zero vertical and horizontal wind. At the top the horizontal velocity is commonly set to the geostrophic wind and the vertical velocity is zero. However, we have turned the geostrophic wind off (this is a homogeneous horizontal pressure gradient): $(u_g, v_g) = (0, 0)$. Of course, due to the differences in surface heating, local pressure gradients will still develop. For potential temperature and humidity we have Neumann conditions at the lower boundary (given by the surface fluxes) and also at the top boundary (where the flux is given by the lapse rate at initialization). The domain is initialized with constant profiles for the velocity (equal to the geostrophic wind for $x$ and $y$ and zero for the vertical velocity). The initial profiles are homogeneous in $x$ and $y$ and for potential temperature ($\theta$) it reads

$$\theta(z) = 300\,\mathrm{K} - 0.01\,\mathrm{K/m} \times (z - 1\,\mathrm{km}) \times H(z - 1\,\mathrm{km}) \tag{1}$$

with $H(\cdot)$ the Heaviside function. The top of the domain is situated within a stable inversion layer, which prevents that the turbulence within the boundary-layer is influenced by the vertical domain size. In the lateral dimensions the domain is about 3 to 5 times the boundary-layer depth. For the vertical velocity we have added an a very small subsidence term (leading to a vertical pressure gradient in the equations) for heights above 1 km to counteract the destabilizing influence of the surface heat flux, with the subsidence velocity $w_s = -0.00003\,\mathrm{s}^{-1} \times (z - 1\,\mathrm{km})$ for all simulations. The data are extracted for four hours after two hours of spin-up time. The data are extracted for four hours after two hours of spin-up time. For each hour a data point is collected by averaging over virtual measurements sampled at every second. As our focus lies on the influence of the surface characteristics, we concentrate in the present study on the wind circulations purely generated by the surface heat flux, without complicating the analysis with additional synoptic drivers such as a geostrophic wind.

2. The simulations: all the information needed to reproduce the simulations exactly should be given. For example, the exact values of initial and boundary conditions of all variables, the strength of the inversion and the subsidence, etc. Im still confused about how many simulations were run. Im assuming it was two, one for the kilometer and another for the hectometer case. If so, the information in Tables 2 and 3 are confusing. Does the word cases mean patches? What are the ranges in ABL height and Obukhov length, are they in time or space? Are these ranges resonable?

*We added the information about the initial and boundary conditions (see above). We clarified the distinction between patch and case in the methods section. In brief, the word "cases" means the different simulations within one suite with varying parameters. The words patches means the different surface types, the number of patches depends on the surface length scales of the particular simulation, which depends on the case. The total number of simulations was 288, one suite of 144 simulations for the kilometer scale heterogeneities and another suite of 144 simulations for the hectometer scale heterogeneities. We have stated this more clearly within the text. The ranges indicate the spread between these 144 simulations, as such neither in time or physical space, but a spread in the "parameter space" of the suite. The ranges are reasonable for free convection.*

**Excerpts from methods section:**

In Fig. 1 we plot an example of a synthetic surface heat flux as in (1) creating eight patches on the surface with four different values for the surface sensible heat flux. The number of patches depends on the length scale of the heterogeneity.

The main aim of this parameter study is to find out the response of virtual towers when the surface parameters are varied, and for this reason we create a suite of simulations where each simulated case has another combination of the surface parameters.

3. p. 6, l. 2122: the Gauß-Ostrogradski theorem has been used to reformulate a divergence within the control volume as a surface term, please be explicit on what was done.

*We added it as a footnote explaining this standard theorem in vector analysis and differential geometry. Further information can be found in e.g. Methods of Theoretical Physics by Morse and Feschbach. For more information upon the mathematical terminology, see e.g. Rudin's Principles of Mathematical Analysis.*

**In the manuscript:**

Where possible, the Gauß -Ostrogradski theorem [1] has been used to reformulate a divergence within the control volume as a surface term. Due to the choice of a cuboid aligned with the coordinate system for the control volume, the control volume energy balance (4) simplifies further because only the velocity components perpendicular to the faces remain.

[1] The Gauß -Ostrogradski theorem or "divergence theorem" is a special case of the Stokes-Cartan theorem in differential geometry. For our purposes, we also restrict ourselves to three-dimensional space. We consider a compact volume $V$ with a piecewise smooth boundary $S$. If $\mathbf{F}$ is a continuously differentiable vector field defined on a neighborhood of $V$, then:

$$\int_V (\nabla \cdot \mathbf{F})\, \mathrm{dV} = \oint_S \mathbf{F} \cdot \mathbf{dS}. \tag{2}$$

The left side is a volume integral of the divergence of the vector field $\mathbf{F}$ over the volume V, with dV the volume element, and the right side is the surface integral over the boundary of the volume $V$. $\mathbf{dS}$ is the outward pointing unit normal field of the boundary $S = \partial V$ multiplied by the surface element. For our purposes we take $\mathbf{F} = \mathbf{v}\,\theta$ and $V$ is the control volume described in the text.

4. The PCA analysis: in the Methods section, a brief description of the PCA method and how to interpret its results should be given. Right now this is completely left to references, but I think I should be able to understand the technique and the plot overall without having to look in another paper. Also, a more exact description of what was done should be given, including how many and which variables were used, which equation or software, etc.

*We expanded the paragraph on the PCA method and its interpretation.*

**In manuscript:**

For investigating the response of the virtual tower measurements to the changes in the parameters, and for investigating the correlation between the measured variables, a principle component analysis (PCA) is applied. PCA relies on the singular value decomposition (SVD) of the data matrix, which consists of the data points for each of the data variables. Through SVD, the data matrix decomposes into the matrix of the left eigenvectors, a diagonal matrix with the singular values and a matrix with the right eigenvectors. The singular values are ordered by their magnitude, because the square of each singular value is the variance of the data explained by its corresponding eigenvectors. Hence the first eigenvectors with the largest singular values represent the principal components that explain the largest fractions of the data variability. We will present many of our results in correlation biplots introduced by Gabriel (1971, 1978). Correlation biplots offer a picture of the relationship between the interdependent variables that make up the data matrix through the PCA method. First of all,

for a correlation biplot, each variable is centered around its mean and normalized by its standard deviation. On the normalized data matrix PCA is applied. The data variables are projected into the subspace spanned by the first principal components and then the vectors of the projection within this subspace are plotted in a 2D (or a 3D) biplot, when the two (or three) largest principal components are chosen. In a correlation biplot, the inner product between the variable vectors (hence, the product of their length and the cosine of the angle) directly measures their correlation. The scree plots related to the biplots express how much variance is captured by the principal components by plotting the (relative) variance explained by the principal components. The variance explained is a measure for the goodness of the fit. The better the variable can be explained by the first two principal components, the longer the length of the vector of the variable in a two-dimensional correlation biplot, which is at most unity, indicated by the unit circle. For a pedagogical description of biplots see e.g. Greenacre (2010).

For producing the biplots we have made use of Python3, combining our own routines with standard packages. The correlation matrix contains exactly the same variables that are plotted in the biplot.

5. Simulation results: the simulation results should be presented and discussed before presenting the statistics. For example, how do the spatial fields of temperature and heat flux look like, and where do the towers rest in this field? Some time series at the tower place, to see what the towers are measuring and what are the scales of motion in it. How do they compare with the scales of heterogeneity? How much of the fluxes are resolved compared to the sub-grid scale? How realistic are these simulations? A thorough discussion of the simulation is definitely needed, as it would help to discuss the physics of the results presented later.

*To the discussion we added an analysis of the simulation results: where the updrafts are concentrated, the dependence of the friction velocity on the heterogeneity amplitude, and a comparison with the homogeneous reference case. Our 10 m grid spacing is more than sufficient to capture the turbulence responsible for the motions in the boundary-layer (the heterogeneity is large compared to the grid spacing) and therefore our simulations are realistic. Of course the measurement height is higher than would be ideal and the control volume method is an approximation to the tower measurements, but these are approximations out of computational grounds. To speak of LES in the first place most of the flux has to be resolved at the subgrid-scale flux is only important at the lowest grid points, which is why we evaluate the energy balance closure at the fifth grid point (50 m).*

**Added in discussion:** 3.1 Circulation patterns in heterogeneous terrain

6. Discussion: there are three distinct physical phenomenon that could be impacting the residual: transport of the mean field, transport of the fluctuating field, and storage. What are the physics involved in each process, how is the simulation capturing them,

and how do they look like in the simulation? Is it realistic to look into the advection effect, for example, in a simulation that has no mean streamwise advection? What happened to the vertical and horizontal dispersive fluxes?

*The simulation captures all turbulent processes on scales larger than the gridsize and smaller than the boundary layer depth (when the horizontal extent is sufficiently large). Transport of the mean field is well captured, transport of the fluctuating field is captured up to the grid cutoff scale (in our simulations this would roughly correspond to resolution of the turbulent fields up to $1\,\mathrm{Hz}$). The storage fluctuations are captured up to the tiny changes faster than $1\,\mathrm{Hz}$ but its average behaviour is well captured. Due to the boundary conditions (and the smaller eddy sizes there) at the lower surface in the first grid points the turbulence is not completely resolved. This is exactly the reason why we study the energy balance closure at the fifth grid point.*

*Even though the simulation has no mean streamwise advection, there is still locally advection. This is realistic, because in reality near thermal updrafts there is convergence near the lower surface and hence also net advection due to continuity of the air fluid (at least within the window that the thermal resides at that location, but Kanda et al (2004) showed that this leads to net effects for half hour windows). For our heterogeneous simulation the thermals are preferentially attached to the hotter patches, leading to advection effects on longer timescales. There is no contradiction here, neither does it make the simulation unrealistic. The vertical dispersive flux is the turbulent flux at the measurement height. The horizontal dispersive fluxes is taken into account by considering the flux-divergence.*

*In short, locally there is still advection even in free convection, as the convection cells create their own circulation. We focus on free convection for two reasons: one it disentangles the "purely heterogeneous" effect from the meteorology, second, in the literature it is found that meteorological conditions towards pure convection lead to stronger imbalances.*

7. p. 7, l. 19: what is the available energy? It is the reference value in the results section, but there is no definition of it. It is only explained in the Fig. 2 caption, but it should be clear in the text too.

   *We have added it to the main text.*

   **Changes in manuscript:** for our model setup the available energy is equal to the surface flux

8. p. 7, l. 19: please make it clear if the advection term includes both horizontal and vertical advection.

   *It does. We added this explicitly in the text.*

   **In manuscript:** this means that our advection term is the sum of the horizontal and vertical advection by the mean flow

9. p. 7, l. 29‑31: We notice that most towers show the typical underestimation of the energy balance, except for the tower located at the warmest spot where there is an updraft. How do you know it is an updraft? Is it always an updraft? On average? It is mentioned in the abstract that updraft and downdraft positions were chosen for the towers, but this is not showed or discussed in the paper. Do constant regions of updraft/downdraft exist in the simulation or in reality?

   *With "updraft" we do not necessarily mean the central region of a thermal, but more generally the existence of $\bar{w} > 0$. In real complex terrain there are certainly areas that have preferential updrafts, e.g. above a mountain range during summer days next to the relatively flat foothills (this phenomenon is called "Alpine pumping") or on a smaller scale, in mixed terrain consisting of darker forests and cooler lakes paragliders can make use of the updrafts above the forests in summer (pers. comm.). Therefore we are not surprized to see preferential updrafts above the hotter patches.*

   **Added in the introduction:** Persistent updrafts and downdrafts tied to the landscape heterogeneity have been found e.g. by Mauder et al (2008) during the 2008 Ottawa field campaign.

   *We added a spatial analysis of the updrafts in the results section 3.1*

10. p. 7, l. 30‑31: please explain better physically the causes of a negative/positive residual, and why there is a negative residual where there is an updraft.

    *The negative residual (turbulent flux larger than the surface flux) appears because there is a net advection of sensible heat from the downdraft region towards the updraft region. A complete physical understanding of the coupling between the residual and the wind field is beyond our present capacities and this paper is one step to understanding the linking. We can however give a few arguments, which we added to the discussion.*

    We present some arguments why the regions with updrafts have better closure. Banerjee et al (2017) investigated the dependence of the aerodynamic resistance on the atmospheric stability for homogeneous terrain. As a consequence a surface with a higher surface heat flux is more efficient in transporting away this surface flux. Therefore, one hypothesis is that when a patch with higher surface flux is coupled to a patch with lower surface flux in heterogeneous terrain, the patch with the higher surface flux transports part of the surface flux of the patch with the lower surface flux, due to its higher efficiency, leading to a net advection of sensible heat from the downdraft region to the updraft region. Another hypothesis is that the shape of the cellular convection cells matters: the updrafts cover a smaller area than the downdrafts. Therefore, as the turbulence structures move across the towers, above a region with preferential updrafts, the likelihood of sampling both the updrafts and downdrafts is higher than above a region with preferential downdrafts.

11. p. 8, l. 13‑14: In the left panel of Fig. 4 we note that the normalized flux-divergence correlates rather well to the normalized turbulent flux, when we look at their average behavior at each tower. What does normalized mean? If it is the normalized by the

available energy at their respective location mentioned in the caption of Fig. 2, it should be mentioned in the text too.

*This was only mentioned for Fig. 2 at the beginning of the section. We now repeat this in the paragraphs for Fig. 3.*

**In the manuscript:** Let us however take a closer look in Fig. 3, where the flux-divergence and advection by the mean flow, resp. are plotted against the energy balance ratio. As in Fig. 2 flux-divergence and advection are normalized by the available energy (i.e. the surface flux in our settings).

12. p. 8, l. 2223: flux-divergence and advection separately correlate well with energy balance ratio and consequently also with each other. What does that mean physically? Does it makes sense that these two processes are correlated in the simulation? What is the implication of this for the imbalance observed in the real case?

*It means, e.g. for the kilometer scale, that a larger advection (due to higher mean temperature or higher wind speed) is coupled to less correlation between the temperature and the velocity fluctuations. The precise mechanisms of this phenomenon are still unclear. We are confident that when the simulation shows such correlation, this should be there in reality too. For turbulence of scales larger than 10 m (the grid spacing) our LES is a good model for turbulence of the same order in reality. One of the implications would be that the EBR can be explained by the advection or flux-divergence only, because the latter two are also well correlated.*

**Added in the conclusions:** Remarkably, flux-divergence and advection by the mean flow correlate separately very well with the energy balance ratio, which implies that the EBR can be explained by the advection or flux-divergence only, as the latter two are well correlated among themselves.

13. p. 8, l. 2426: "Finally, we want to remark that, due to computational constraints, the virtual measurement height in our simulations lies at 50 meters, which is an order of magnitude larger than the typical tower height over short vegetation with comparable surface roughness. This means that our findings for virtual EC towers cannot be directly transferred to real eddy-covariance towers." Why not compare the residual observed in the simulation as a function of height, including the lower points? If I understood correctly, the residual term is estimated at a given height, so it does not need the control volume approach. If so, you can see if the results at 50m are similar at 10 m, and if the conclusions could be extrapolated. Without this extrapolation, I see little usefulness in the conclusions obtained here.

*We need the control volume in order to be able to estimate advection and flux-divergence more accurately. We need a minimal number of grid points in the vertical because at the lowest grid level, the turbulence is parameterized by MOST. Extrapolation of the residual from 10 m height would only make sense if we resolve the simulation with O(1 m) resolution, and then we could study the height dependence. We've already done a separate study on the variation with height of the energy balance*

*closure for smaller grid sizes in homogeneous terrain, but these simulations were too computationally expensive to set up for the two suites of 288 simulations in total, especially because the presence of the heterogeneity calls for a larger domain in the horizontal. Finally, the residual is calculated as the difference between turbulent flux at the top and the surface flux (we're actually considering a local area average). On its own it does not need the control volume method but we need the control volume method to quantify advection and flux-divergence. However, the problem remains that the turbulence in the first grid points is not completely resolved, but this is a general issue of LES due to the lower boundary condition.*

**In the conclusions:** By means of a control volume approach, we decomposed the modeled surface energy budget to highlight its partitioning, and we have shown that the modeled energy balance ratio exhibits values that are found in field experiments. In addition, this approach allows us to investigate the energy balance closure in two-dimensional complex terrain.

14. p. 8, l. 3233: "the towers in the center of the patches even behave in the opposite manner when the kilometer and hectometer scales are compared". This needs to be better investigated and discussed physically. What can be causing this?

    *We refer to comment 16.*

15. Discussion: I think it is important to discuss the differences when there is a small residual due to low values of all other fluxes, or due to their canceling effect. What is likely to be happening in field experiments?

    *For field experiments in general we expect that this will be something site-specific. We do notice here and also in Eder et al 2015 that the flux-divergence and the advection act oppositely, but for different sites with different type of heterogeneity or different synoptic or radiative conditions the residual could also be present because all fluxes are small. There is a multitude of factors that play a role and this paper only focuses on strongly convective conditions to free convection for chessboard shaped heterogeneity of 2 times kilometer scale and 2 times hectometer scale, which already required 288 simulations.*

16. p. 9, l. 4–7: "The likely cause for the different behavior between the two scales of heterogeneity would be the blending of the hectometer landscape heterogeneity due to the virtual tower heights of 50 meter. For the surface heterogeneity of $O(10^2$ m) the flux footprint of each of the towers can cover several of the surface patches, regardless of the type of tower." I agree this is likely the cause of difference, but how does this make the relation between flux-divergence and EBR opposite, or the relation between the residual and the surface flux opposite?

    *One way to investigate this would be to simulate additional length scales interpolating between our current heterogeneity lengths and investigate at what point the behaviour switches. However, our periodic boundary condition in the horizontal places strong*

*constraints on the possible heterogeneity lengths (the heterogeneity length has to be a divisor of the domain size). Furthermore, it would have added a significant number of additional simulations.*

17. PCA results: from the very little I know about PCA analysis, I think the results obtained here are not useful. I believe it is useful to look at correlation biplots when most of the variance are explained by the first two PCs (something around 90%), not the 60% found here. As explained by Greenacre (2010) (your own reference on the topic), PCA correlation biplots are useful when there is a clear separation in the scree plot between the first two PCs and the rest (called elbow), which is definitely not the case here. In your data, the third PC is almost as important as the second, and it is not taken into account. In addition, the EBR (the variable you want to explain) is the one with less representation by the first two PCs among the variables in the correlation biplot, being much less than 50% representation in the hectometer case. I dont know which analysis should be done instead, but I think PCA is not the one.

*Yes, we know that biplots are more useful when the variance explained of the first two PC's is higher than 60% and the elbow is more pronounced. We do not claim otherwise in the text, but we made it clearer now. Nevertheless, even when the variance explained is lower than 60%, we can still make a biplot. The PCA is always done with all the principal components, the biplot is just the figure of the first two PC's. We chose for the correlation biplot precisely because the unit circle gives guidance as to how well the variance is captured by the first two components. The arrows which are close to the unit circle in the correlation biplot are still well explained by the first principal components. The residual variance from the latter PC's then explains the short arrows in the biplot. Alternatively, we could yield tables with the correlation coefficients, which is the same information as in the biplot, just not visual. The variance explained concerns all the variables, but we are mainly interested in the EBR. Anyway, we also think it's still meaningful to show that we do not see enough correlation from a linear analysis. We also think it is more instructive to give a graphical representation of the correlations. The PCA is only a linear analysis and therefore a higher order scheme might help, but it would introduce a lot of dimensional coefficients and it is sometimes said that "with a fifth order spline one can fit an elephant". We think it is necessary to a come to a deeper physical understanding in order to be able to fit a higher-dimensional model with a minimal number of parameters that could be used in field-experiments.*

**Added in the results:** *Despite the variance explained of around 60%, plotting the correlation biplot is still possible, it remains a graphical representation of the correlation between the variable vectors, even though it shows that the variance in those variables cannot be simply explained by the first two principal components. Furthermore, the linear correlation analysis in the biplots is still useful in a few other respects: for one, it shows that EBR does not linearly depend on easily measured*

characteristics.

18. Conclusions: ignoring the results from the PCA, the only conclusion from this work is that this type of heterogeneity in the surface flux generates a difference between the turbulent flux measured at 50m and the surface flux itself. This is an interesting information, but given the potential of these simulations, I believe that much more can be obtained. I would like to see some results and conclusions that could help improve or understand the closure problem in field experiments. The idea of finding parameters measurable in the field that correlate well with the residual is a good approach. But another technique should be used to find the right parameters. After finding them, I suggest that a model for the residual should be developed and tested against the LES results. Naturally, in reality things are not equal to the LES, but this would gives us a place to start. When combined with a thorough discussion of the simulation itself, it would be easier to extrapolate the results and conclusions to the real case. Without it, I dont see any significant contribution to the field.

*In addition to a more extensive physical discussion of the simulation output (see below for additions) To find the right parameters that can be used in a model will require additional physical analysis because linear correlation analysis a la PCA has not provided us with clear insights. There is a simple model by Huang et al (2008) but it focuses on the mixed layer and its applicability near the surface is limited (its fit is based on the mixed layer and some of the fitting functions are undefined near the surface). We think that our paper with its investigation of the energy balance closure problem at a coarser resolution has its merits, by decomposing the imbalance as found by the LES simulations in a way that allows us to quantify the individual components, by showing it is in the right order of magnitude of field experiments (though the setting is simplified but we vary the parameter space that determines the surface heterogeneity) and from the simplified setting trying to find the parameters influencing the EBC over a range of the parameter space. With respect to "given the potential of the simulations", we think that we got most of the potential out of it, at least with respect to the goal of the paper. The high correlation between advection, flux divergence and the energy balance residual is also an important conclusion.*

**Added in PCA discussion:** In addition, friction velocity and boundary-layer depth cluster together separately, as do the normalized flux-divergence and advection. Although we might have supposed that higher boundary-layer heights will arise if patches are present with vigorous surface heating, however we found that $u^*$ decreased with stronger surface heterogeneity. Closer analysis reveals that the highest boundary layer heights are obtained when the heterogeneity amplitudes are smaller and the domain is more homogeneous. Hence the former clustering can be explained because in our scenario with varying heterogeneity amplitudes the highest boundary-layer height and larger $u^*$ are both obtained for smaller heterogeneity amplitudes.

**Added in PCA discussion:** In the literature (e.g. Stoy et al 2013, Eder et al 2015a) a correlation between friction velocity and energy balance closure has been found: a

high friction velocity leads to a smaller residual. Typically, a higher friction velocity is correlated to smaller atmospheric instability, and hence roll-like convection instead of cellular convection. Maronga and Raasch (2013) found that boundary-layer rolls "smear out" the surface heterogeneity, leading to an effective surface that looks less heterogeneous, which has been related to a higher EBR (Mauder et al 2007, Stoy et al 2013). Therefore a possible cause for the present low correlation of $u^*$ with the EBR, could be our range of the stability parameter. For the free convective cases considered here the stability parameter lies below the range where the friction velocity has a high correlation with EBR.

**Added in conclusions:** By means of a control volume approach, we have decomposed the modeled surface energy budget to highlight its partitioning, and we have shown that the modeled energy 20 balance ratio exhibits values that are found in field experiments. In addition, this approach allows us to investigate the energy balance closure in two-dimensional complex terrain.

**1.3   Technical corrections**

- Abstract: there is too much introduction information in this abstract. It is also clear from it that there is no significant contribution from this work. After improving the results and conclusion, the abstract should focus more on them.

  *We suppressed the introductory information in the abstract and focused on the conclusions of the article.*

  **New abstract:**The imbalance of the surface energy budget in eddy-covariance measurements is still an unsolved problem. A possible cause is the presence of land surface heterogeneity, which affects the boundary-layer turbulence. To investigate the impact of surface variables on the partitioning of the energy budget of flux measurements in the surface layer under convective conditions, we set up a systematic parameter study by means of large-eddy simulation. For the study we use a virtual control volume approach, which allows the determination of advection by the mean flow, flux-divergence and storage terms of the energy budget at the virtual measurement site, in addition to the standard turbulent flux. We focus on the heterogeneity of the surface fluxes and keep the topography flat. The surface fluxes vary locally in intensity and these patches have different length scales. Intensity and length scales can vary for the two horizontal dimensions but follow an idealized chessboard pattern. Our main focus lies on surface heterogeneity of the kilometer scale, and one order of magnitude smaller. For these two length scales, we investigate the average response of the fluxes at a number of virtual towers, when varying the heterogeneity length within the length scale and when varying the contrast between the different patches. For each simulation, virtual measurement towers were positioned at functionally different positions (e.g. downdraft region, updraft region, at border between domains, etc.). As the storage term is always small, the non-closure is given by the sum of the advection by the mean flow and the flux-divergence. Remarkably, the

missing flux can be described by either the advection by the mean flow or the flux-divergence separately, because the latter two have a high correlation with each other. For kilometer scale heterogeneity, we notice a clear dependence of the updrafts and downdrafts on the surface heterogeneity, and likewise, we also see a dependence of the energy partitioning on the tower location. For the hectometer scale we do not notice such a clear dependence. Finally, we seek correlators for the energy balance ratio and the energy residual in the simulations. Besides the expected correlation with measurable atmospheric quantities such as the friction velocity, boundary-layer depth and temperature and moisture gradients, we have also found an unexpected correlation with the temperature difference between sonic temperature and surface temperature. The correlation with the friction velocity is less pronounced than previously found, but this is likely due to our concentration on effectively strongly to freely convective conditions.

- p. 4, l. 17: "we have added a very small moisture flux" why? why not make it zero?

  *We did not choose to run dry simulations, as it is sometimes viewed as less realistic. Of course, a simulation with a very small moisture flux approximates the results of a dry simulation. In any case, we wanted to concentrate on the partitioning of the sensible heat flux before adding a significant latent heat flux.*

- Figure 1: missing unit of color plot (surface heat flux)

  *It's in* $\mathrm{Kms}^{-1}$ *(added)*

- Equation (3): a sketch of the fluxes and a figure showing where in the 50 by 50m box each term is being calculated would be useful.

  *To avoid copyright issues we prefer to refer to Eder et al (2015).*

- p. 7, l. 30: include text: "most towers show the typical underestimation of the energy balance (i.e., positive energy balance residual)"

  *Done*

- p. 8, l. 12: what is "resp."?

  *Respectively.*

- p. 8, l. 12, 13: it should be Fig. 3 instead of Fig. 4

  *Thanks, corrected.*

- p. 8, l. 20: opposite slope? They look the same to me...

  *Yes, but the text is really correct because we have unfortunately made an error while editing the figures for the approved submission. On demand of reviewer 2 in the quick submission we had removed some plots, and in this action we accidentally plotted figure 5 on the location of figure 3 as well (with two times the caption of figure three, to make matters worse). We updated with the correct figure, where the slope*

*is opposite. In the original submission for the quick review we had the correct plot. The latest figure also has the different markers for each tower type, compared to the plots in the quick review submission.*

- p. 8, l. 31: since you are comparing Fig. 2 with Fig. 4, why not make them one single plot? It makes it easier to compare.

  *It might be easier to compare, but we think that the details come out better in two separate plots.*

- Fig. 3 and Fig. 5 are equal, one of them is wrong (I guess it is Fig. 3, based on the text)

  *Indeed, see above. In the quick review submission we had the correct figure, which we repeat for this revised submission (with individual tower markers).*

- p. 9, l. 2: Fig. 4 should be Fig. 3

  *Thanks, corrected.*

---

## Author Comment (AC2) · 30 Dec 2017

**Reply to the reviewers**

Frederik De Roo and Matthias Mauder

December 2017

*We thank the reviewers for their careful reading and constructive comments. We think that our implementation of the comments has improved the discussion paper and made it clearer for a wider audience.*

*Our responses are in italic, and our changes in the manuscript are straight. We have used a brown color for the part of the comments of reviewer 2 that we already incorporated in the original Discussion paper, and a blue response for the recent changes.*

**1   Reviewer 2**

**1.1   Major comments**

1. Energy balance (Eq. 3)

   *First of all, we made a typo when typing the storage term (not in our data processing) but other than that the main misunderstanding seems to come from our application of the divergence theorem. Applying Gauss-Ostrogradski (or Stokes-Cartan if need be) is a common procedure in a mathematical treatment of fluid dynamics and is textbook material, see e.g. "Fundamentals of Momentum, Heat and Mass Transfer" by Welty et al.*

   I am struggling with this equation due to several reasons. First, the dimensions do not fit. A flux should be given in K m/s. The storge term, however, is the integral of d/dt over all three dimensions in space so that the resulting dimension is K / s * m3 = K m/s * m2, which is NOT a flux. The equation thus can't be correct (this applies to the original equation in Eder et al. 2015 as well).

   *Thanks for catching our typo in the equation, the dimensions of course work out as they should. We added the normalization factor for the volume averaging (so the integral turns into an average). The volume integral is normalized by the surface area, which is the same as taking the average line integral over z, the way it is presented now. We already corrected this in the discussion paper.*

   Second, the authors refer to the second right hand side term as the flux divergence. That is not correct. The divergence involves the derivation in space, but here only

a sum (difference) is calculated. Using the divergence, the equation would start to make sense, as the dimension of a flux divergence is K/s, which fits to a storage term (d/dt).

*That term is the average flux-divergence over the volume, it's just reformulated into a difference. The flux divergence at a point indeed has a spatial derivative, but after integration over the control volume (and normalization) the derivative can be removed.*

Third, to me it would make more sense to put storage on the left hand side and the rest on the right hand side.

*Whether it's a plus sign on the right or a minus sign on the left it's a matter of taste. The surface flux is our constraint, we put all other terms on the right where they retain a plus sign.*

Fourth, why is the storage term integrated over a volume? Which volume? Shouldn't ALL terms be integrated over the entire control volumes? Is it really an integral or an averaging?

*Which volume: "the" volume of the control volume. All terms are integrated over the entire control volume, but the storage term is the only term that cannot be (analytically) reformulated into a surface term because it has no (spatial) divergence in front.*

Moreover, you name the $< w >< \theta >$ terms "vertical advection" and so on. But a turbulent flux is also advective, so this is not correct language. You mean the advection by the mean flow, which the flux divergence represents the advection by the turbulent flow?!

*We replaced the shorthand "advection" by "advection by the main flow". Flux-divergence is a standard term so we keep it.*

Lastly, The summation over the lateral fluxes seems to work, but I expect the same summation over the vertical fluxes, i.e., $\overline{w'\theta'} + H$, but even in the equation it seems to be a difference and not a sum ( $\overline{w'\theta'} - H$ if you put all terms on one side). In Summary, the equation is very difficult to understand, and reading the Eder et al. paper does not really contribute to the understanding.

*It has to be a difference in order to close the energy balance over the control volume. The convention is that a positive H is directed upward. Imagine that there were no advection and no storage, then the turbulent flux should equal the surface flux. Hence $\overline{w'\theta'} - H = 0$.*

Finally, I am also not sure that you can apply your averaging procedure for such a large control volume. Temperature, e.g. has a logarithmic profile near the surface, so you must be careful when integrating the storage term over all control volumes.

*While it is in principle true that we have to be careful when integrating a logarithmic profile by adding the interpolated values, in our convective simulations the log*

*profile is only in the first gridpoint (by applying MOST) and due to the smallness of the roughness length compared to the half-height of the gridsize the argument of the logarithm is very large, where the behaviour of the log can be well approximated by a linear function, leading to a good approximation of the integral by the sum of the gridpoint values. Therefore the error is quite small. Anyway, the storage term is small on its own, so this is only a small correction on a small term.*

**In the manuscript:**

Where possible, the Gauß-Ostrogradski theorem [1] has been used to reformulate a divergence within the control volume as a surface term. Due to the choice of a cuboid aligned with the coordinate system for the control volume, the control volume energy balance (4) simplifies further because only the velocity components perpendicular to the faces remain.

[1] The Gauß– Ostrogradski theorem or "divergence theorem" is a special case of the Stokes-Cartan theorem in differential geometry. For our purposes, we also restrict ourselves to three-dimensional space. We consider a compact volume $V$ with a piecewise smooth boundary $S$. If $\mathbf{F}$ is a continuously differentiable vector field defined on a neighborhood of $V$, then:

$$\int_V (\nabla \cdot \mathbf{F}) \, \mathrm{dV} = \oint_S \mathbf{F} \cdot \mathbf{dS} \,. \tag{1}$$

The left side is a volume integral of the divergence of the vector field $\mathbf{F}$ over the volume V, with dV the volume element, and the right side is the surface integral over the boundary of the volume $V$. $\mathbf{dS}$ is the outward pointing unit normal field of the boundary $S = \partial V$ multiplied by the surface element. For our purposes we take $\mathbf{F} = \mathbf{v}\,\theta$ and $V$ is the control volume described in the text.

2. Homogeneous control run From Fig. 2 it becomes obvious that the EC flux cannot make up for the available energy. I think that is the main point of the paper. It is shown nicely, that advection (of the mean flow) plays an important role, and also flux divergence. You relate this to the heterogeneous surface, but you do not prove it. What you need to do is to add results from control runs with a homogeneous surface. What do you expect to happen? At least advection should go to zero, because all velocity components should go to zero when sufficiently averaged.

*We do have homogeneous control runs, more precisely the cases when the amplitude of the surface flux is zero ($A_x = A_y = 0$): then the terrain is homogeneous because in our setup the heterogeneity can only appear in the spatial variation of the surface flux. We would like to stress that also in homogeneous terrain the energy balance is not necessarily closed, see e.g. Kanda et al (2004), because turbulence structures can exist for longer than e.g. half an hour. We agree that when "sufficient averaging" is applied in homogeneous terrain (assuming stationarity of the turbulence) the EBR should be one.*

**In manuscript: 3.1 Circulation patterns in heterogeneous terrain**

3. Measurement height I do understand why you did your virtual measurements at an elevated height because of a relatively coarse grid spacing and you need 5 grid volumens. However, I could not find the grid spacing in the text, so I had to infer it from the table.The main point, however, is and you state that yourself that EC measurements are performed much closer to the surface (2 m) so that it is completely unknown whether your results are transferable to 2 m height. Maronga and Raasch (2003) showed, e.g., that the effect of secondary circulations (heterogeneity effect) tends to zero close to the surface and is most pronounced at upper levels. To me it remains unclear what the benefit of your study is to the research community.

   *We added the grid spacing and the other important dials of Table 1 in the main text (see also the first comment of reviewer 1).*

   **In the methods:** Relevant parameters of the simulation setup are summarized in Table 1, the grid spacing is $10\,m$ in all three dimensions and the domain size is $6 \times 6$ square kilometers in the horizontal, and $2.4\,km$ in the vertical. The boundary conditions of the simulations are periodic in the lateral dimensions.

   *While we strive towards simulating real EC measurements performed at 2 m height (which is extremely costly computationally because it requires less than 0.5 m resolution, without suitable model manipulations that we are working on) these simulations are a first step towards that goal, at the time we used the computing time available to us to produce the 288 runs. This article provides a method that can be repeated for higher grid resolution. Though the domain averaged energy balance of Maronga and Raasch (2013) tends to zero close to the surface, we do not think it means that it's absent (in fact, in reality the non-closure is there at 2 m) and we are still investigating this. There are some issues related to high-resolution LES near the surface, not only related to the computational demand, so we think this problem merits additional research. One of the major benefits of our approach is that we can study the energy balance closure locally in 2D complex terrain.*

   **Added in the conclusions:** By means of a control volume approach, we decomposed the modeled surface energy budget to highlight its partitioning, and we have shown that the modeled energy balance ratio exhibits values that are found in field experiments. In addition, this approach allows us to investigate the energy balance closure in two-dimensional complex terrain.

4. Figures and length I suggest to shorten the paper and stick to the main message. Not all figures are needed I think. Especially the special plots are difficult to read given the rather simple message you want to convey. By the way, figure 1 could be plotted nicer.

   *We combined plots 2 and 3 into plot 2 (new) and 5 and 6 into plot 4 (new). We inserted different markers for each tower in figure 1.*

5. Language You should be more careful with language editing before submission. I understand that correct English is an issue for non-native speakers, but please avoid

wrong syntax and incorrect formatting. Examples: P1 L4: there is an extra "was", at several points you use the phrase "in function of", which to my knowledge simply does not exist, P7 L14: "Table table:2", wrong formatting. Referees spend their rare time for reading your manuscript without having any benefit from it. Beeing confronted with such carelessness suggests that the authors did not read their own manuscript carefully and this does not really motivate to review your manuscript in-depth. See my detailed language corrections below.

*We apologize for the typos that we didn't catch beforehand and thank you for your time and efforts. We already incorporated these corrections in the Discussion paper.*

**1.2   Minor comments**

1. P2 L17: Why is there a correlation between the friction velocity and the energy balance closure? Also: there is nothing as a good correlation. It has a value, and can be possible regarded as high.

   *Corrected: good → high*

   **In the results:** In the literature (e.g. Stoy et al.,2013; Eder et al., 2015b) a correlation between friction velocity and energy balance closure has been found: a high friction velocity leads to a smaller residual. Typically, a higher friction velocity is correlated to smaller atmospheric instability, and hence roll-like convection instead of cellular convection. Maronga and Raasch (2013) found that boundary-layer rolls "smear out" the surface heterogeneity, leading to an effective surface that looks less heterogeneous, which has been related to a higher EBR (Mauder et al., 2007; Stoy et al., 2013). Therefore a possible cause for the present low correlation of $u^*$ with the EBR, could be our range of the stability parameter. For the free convective cases considered here the stability parameter lies below the range where the friction velocity has a high correlation with EBR.

2. P2 L19: You talk about secondary circulations by self-organization, but I am not sure this contradicts the definition of a secondary circulation. Imagine a self-organized flow (give an example!), then what is the difference between primary and secondary circulation? I consider the hexagonal patterns for instance as the typical case of self-organization in the CBL, but that is what we call the primary circulation.

   *We agree that the hexagonal patters should be called primary circulation in the homogeneous case. The secondary circulation is the additional circulation due to the presence of the heterogeneity (insofar it is a small effects that can be superimposed upon the primary circulation, as the NS equations are nonlinear). However, in our case the hexagonal patterns arising from the convective conditions are preferentially tied to the landscape heterogeneity in the case of O(km) heterogeneity, so the primary and secondary circulation become indistinguishable.*

   **In the introduction:** In the case of cellular convection in heterogeneous terrain

the distinction between the primary and the secondary circulation becomes blurred, when the convection cells are tied to the landscape heterogeneity.

3. P2 L29: Secondary circulations do not decay to zero at the surface! They are simply not there.

   **Rewritten:** "they are not present at the ground level".

4. P3 L5: Why is w=0 at first grid level?

   *The lowest level (I would prefer to call it level zero) is the level where the boundary condition is applied. By constructing in the LES, there is no vertical velocity at the ground level which is supposed to be a hard boundary, not a movable or permeable membrane. We summarize this as a "rigid no-slip boundary condition" (added in the text) which implies zero values for the horizontal and vertical wind components.*

   **In manuscript:** Firstly, from continuity we indeed expect no vertical meso-scale transport by advection with the mean flow at the lowest grid point representing the lower surface, since w=0 due to the rigid no-slip boundary, but horizontal flux-divergence plays a role, too.

5. P3 L19: The statement does not make sense. You can't say that a different partitioning of sensible and latent heat fluxes led to a higher Bowen ratio, because that is simply the definition of the Bowen ratio!

   *We didn't intend to say this. The first subclause states the finding (Bowen ratio depends on the scale of the heterogeneity), the second makes it more precise (Bowen ratio is largest for intermediate scales).*

   **Rewritten:** Brunsell et al. (2011) found that the partitioning between latent and sensible heat was affected by the scale of heterogeneity. More precisely, the intermediate scales yielded a higher Bowen ratio.

   *This is one of the results of their article, and we don't want it to be understood as a definition.*

6. P4 L19: What is "strong convection"? All your simulated cases are without mean wind, so you have free convective conditions in all your simulations.

   *We rephrased it as "strongly convective conditions". Taking the Obukhov length as the criterion for the strength of the instability of the CBL we have cases that fit the tag "strongly unstable" of Patton et al (2016) for which they take $L \approx -10$ m. For purely free convection we should at least have $|L| < 1$ which is not satisfied for all cases. Even though we do not have a synoptic background wind, the circulation between the patches generates a nonzero $u^*$ that increases L. When we determine the instability from surface variables ($u^*$ and $\overline{w'T'}$) we cannot call all our cases free convection (in reality we would determine it from the EC tower, so we only have knowledge about the surface variables).*

**In introduction:** To disentangle the influence of the surface heterogeneity from that of the meteorology, we will focus on a set-up of free convection without a synoptic wind (which will effectively lead to strongly to freely convective conditions diagnosed by the virtual towers).

**In the introduction:** In addition, as both the lack of closure and the strength of the circulations are most pronounced for strongly convective conditions, we will likewise focus on (effectively) strongly unstable conditions to free convection, we will likewise focus on strongly unstable conditions to free convection, with the unstability parameter $-z/L$ ranging from 1 to 5000. The $-z/L$ is different from $\infty$ because the convective conditions lead to cellular circulation patterns, which locally induce a friction velocity at the surface, and due to its positiveness, there will also be a horizontally averaged $u^*$ different from zero. From the perspective of the tower measurement, by eddy-covariance measurements alone it cannot be distinguished if a measured $u^*$ follows from the wind aloft or locally from the convection-driven circulation. In addition, the circulation locally leads to advective terms that can influence the energy balance closure: e.g. near an updraft there will be horizontal convergence in the flow field. Even in homogeneous terrain these advective terms can lead to a non-closure of the surface energy budget (e.g. Kanda et al 2004).

*We have also added an analysis of the dependence of the (horizontally averaged) $u^*$ on the amplitude of the surface heterogeneity. In homogeneous terrain a stronger surface heating leads to stronger circulation patterns and therefore the dependence could be complicated. However, it turns out that the domain-averaged $u^*$ drops monotonically with stronger amplitudes of the surface heterogeneity.*

**In the results:** On the topic of circulations driven by a surface conditions that are by design freely convective, we investigate how the domain average of $u^*$ is influenced by the surface heterogeneity. The ratio between the surface flux at the hottest patch and the surface flux at the coolest patch is given by:

$$r = (1 + A_x + A_y + A_x \cdot A_y) \times (1 - A_x - A_y + A_x \cdot A_y)^{-1} . \qquad (2)$$

The horizontal mean of the friction velocity scales very well with the natural logarithm of this ratio,

$$u^* = -0.046 \ln(r) + 0.384 \ , \ R^2 = 0.85 \qquad (3)$$

The remaining spread in $u^*$ does not result from the time stamp or the heterogeneity length scale. The monotonous decrease of $u^*$ in function of the heterogeneity ratio shows that for more homogeneous terrain we will obtain a slightly larger domain averaged $u^*$.

7. P5 L16-31: You state that Neumann conditions are used at the lower boundary, which means you are prescribing surface fluxes. Then, how to you prescribe the surface momentum fluxes? With what values? Somehow you need to take into

account the surface roughness, but for that you will need to use Monin- Obukhov Similarity Theory.

*For the velocity components, Dirichlet conditions are applied. The Neumann conditions are for scalars like potential temperature. There is a roughness length (indicated in table 1) and the first gridpoint is parameterized with MOST. We added a clarification of the model, as demanded by reviewer 1, including a more thorough discussion of our boundary conditions.*

8. P6 L4: How can a tower be homogeneous? The surface around its base can, though.

   *It was shorthand for a tower situated in locally homogeneous terrain. We rephrased this.*

9. P7 L14: There is no dataset in the table.

   *The table summarizes the simulation parameters within that suite. We clarified this in the text and in the table captions.*

10. P7 L31: "worst imbalance of only 69%". I guess you mean closure, not imbalance.

    *Indeed, we meant lowest mean closure, corrected.*

11. P8 L1: finally you mention that you are talking about the "mean advection", but it should read "advection by the mean flow".

    *We replaced "mean advection" everywhere by "advection by the mean flow."*

12. P8 L25: You start the second sentence nearly exactly as you have finished the preceding one. That sounds odd.

    *We removed that sentence.*

13. P8: You discuss the difference between kilometer and hectometer scale heterogeneity; but from previous studies it is known that only those can trigger secondary circulations whose scale is in the order of magnitude of the boundary layer depth. Now I am wondering whether you took this into account or not. In the end, you do not show secondary circulations at all, so it remains a secret whether your findings are related to the scale of the heterogeneity of the ratio of the scale of the heterogeneity to the boundary layer depth. Or do you see local circulations that were not seen in previous studies? If yes, you should show them. You only provide very little evidence here.

    *The references show that also landscape heterogeneity of scale smaller than the boundary layer depth has a signature on the spectrum of the heat flux (e.g. Brunsell et al 2011) and can affect the EBR (Schmid et al 1990). We improved our choice of words and replaced "secondary circulations" by "secondary circulations at the landscape up to the boundary-layer scale". We would expect a dimensionless parameter such as the scale of heterogeneity divided by either $L$ or by $z_i$ (these are the most obvious choices,*

*the former for local circulations, the latter for secondary circulations spanning the entire boundary-layer). Because our heterogeneity length is limited to four values (after all our domain has to be periodic) this is not enough to do the correlation analysis. Nevertheless it is an interesting idea to simulate different heterogeneity lengths (and a larger range of $u^*$) to investigate correlation with $L_{het}/L$ or $L/z_i$ but this study focuses on 288 simulations of strongly to freely convective conditions. We added a subsection on the local circulations.*

**Added in the discussion:** We start our analysis with a discussion of the location of the updrafts and downdrafts in heterogeneous terrain. For this purpose, we concentrate on a few specific cases, more precisely $A_x = A_y = 0.3$ and all four heterogeneity lengths (with $L_x = L_y$). We will take the mean vertical velocity as the simplest proxy for circulation patterns in the boundary layer. In Fig. 2 we plot the time-averaged vertical velocity at the height of the control volumes (50 m). We stress that the structures at 50 m extend into the mixed layer above where the absolute velocities become larger (not shown). The reason for the additional time average (over the complete virtual measurement interval of 4 hours) of the already hourly mean data is to remove the drift of the turbulent structures. We notice that for the heterogeneity lengths of O(km), the motions within the mixed-layer clearly reflect the surface pattern, with updrafts concentrated above the hotter patches and downdrafts above the lower patches in the 3-km heterogeneity length, and a little offset in case of the 1.5-km heterogeneity length. However, the structure of the convective turbulence for both kilometer scale are clearly different from homogeneous control run, where typical cellular convection patterns arise (Schmidt and Schumann 1989), though the hectometer scales are qualitatively rather similar to the homogeneous run. The latter could be a consequence of the blending height. Investigating the heterogeneity lengths of O(hm) with more horizontal detail for the time-averaged $w$, we do not see clear updrafts or downdrafts tied to the surface heterogeneity. However, in this respect it could be interesting to note that some of the hourly mean vertical velocity (without additional time-average) for the O(hm) appear better related the surface structure. Similar results appear for weaker amplitudes and also when $A_x$ is different from $A_y$, in which case the dominant pattern is visible along the direction with the larger amplitude (not shown). We can conclude that circulations are tied to the landscape heterogeneity when it is of O(km). For the O(hm) such a correspondence is unclear. However, the latter could be related to the "coarse" grid resolution and the distance from the ground. Indeed, Mauder et al (2010) found persistent updraft and downdraft regions during the 2008 Ottawa field campaign.

14. P9 L13: Strictly, the friction velocity is zero in free convection as the mean wind is zero, so how can it increase for stronger circulation patterns? You must describe that you treat u* as a local quantity and that the primary circulation creates local wind shear near the surface (if that is what you mean).

    *Yes, we clarified our description. Nevertheless, the circulation pattern that arises*

*from the surface patches leads to a mean nonzero $u^*$ as well, because $u^*$ is always positive. (This does not contradict the mean zero mean wind because that has a direction so the average becomes zero. Hence, locally the flux-gradient assumption can still hold, but it doesn't hold for the area-average of the simulation domain.)*

**In the methods:** In addition, as both the lack of closure and the strength of the circulations are most pronounced for strongly convective conditions, we will like-wise focus on strongly unstable conditions to free convection, we will likewise focus on (effectively) strongly unstable conditions to free convection, with the unstability parameter $-z/L$ ranging from 1 to 5000. The $-z/L$ is different from $\infty$ because the convective conditions lead to cellular circulation patterns, which locally induce a friction velocity at the surface, and due to its positiveness, there will also be a horizontally averaged $u^*$ different from zero. From the perspective of the tower measurement, by eddy-covariance measurements alone it cannot be distinguished if a measured $u^*$ follows from the wind aloft or locally from the convection-driven circulation. In addition, the circulation locally leads to advective terms that can influence the energy balance closure: e.g. near an updraft there will be horizontal convergence in the flow field. Even in homogeneous terrain these advective terms can lead to a non-closure of the surface energy budget (e.g. Kanda et al 2004).

**1.3 Language**

1. P1 L1: pending problem - language.

   *Corrected: unsolved problem*

2. P1 L4: extra was, boundary-layer scale

   *Corrected*

3. P1 L14: order of magnitude instead of decade

   *Corrected*

4. P2 L1: Earth's

   *Corrected*

5. P5 L3: influence on what?

   *On virtual flux measurements. Rewritten.*

6. P5 L18: (and throughout the text): "in function of" → "as a function of"?

   *We have removed all occurrences of "in function of".*

7. P6 L6: what is the "?" for?

   *Missing latex reference. Corrected.*

8. P6 L27,29: Decide: "Hz" or "Hertz"

   *We confused it with the writing of e.g. "1-Hertz sampling" Corrected.*

9. P7 L19: "time steps"

   *Corrected*

10. P7 L24: "we now plot" — rewrite.

    *Rewritten, it's in the same plot as well because we condensed the figures.*

11. Fig. 2: what does "in casu" mean?

    *"In casu" means "in this case". We removed it.*

12. Tab. 2: What does "1.4 ? 2.2" mean?

    *Should have been a dash. Corrected.*

13. Tab. 2: -36,1 should read 36.1,

    *Corrected.*

14. Tab. 2: I'd more like a long-list with all simulations

    *This would yield two tables with 144 rows (6x6x2x2 cases) and 7 columns, which is basically the information in tables 2 and 3.*

15. Tab. 2+3: What is the Boundary-layer height in "short" (1.4 – 2.2), while the Obukhov length is in "long" (From ... to ...)?

    *The Obukhov length is negative for unstable conditions and therefore the dash distracted. Both are now in the long format without a dash.*

---

## Referee Report (RR1)

Review of
**The influence of idealized surface heterogeneity on virtual turbulent flux measurements**

by
Frederik De Roo and Matthias Mauder

January 15, 2018

**General comments:**

This paper is successful in demonstrating the following conclusions: (i) in heterogeneous conditions in the kilometer scale, there is a clear relation between energy budget components and the location of the tower; this relation is less pronounced in the hectometer scale. (ii) In between patches of different surface heat flux, the energy imbalance is larger than in the center of patches. (ii) Storage is less important for the energy imbalance than advection and flux-divergence. (iv) Both advection and flux-divergence correlate well with the energy budget residual, therefore only one of them is needed to explain the residual. (v) In the kilometer scale, the advection is negatively correlated to flux-divergence variation; in the hectometer scale, it is positively correlated.

Although I don't think any of the three major suggestions I mentioned in my previous review were satisfactorily done, namely (i) a thorough discussion of simulation results, (ii) a correction of the correlation analysis, and (iii) a proposition of a model that could be useful in field experiments, I understand that they are more suggestions than corrections, so you should only take the suggestions you see fit. I believe the manuscript now has most of the information needed to sustain the aforementioned conclusions and only need minor revisions, although I still believe that better analyses would increase the impact of the manuscript significantly. The topic is important and the simulations are very interesting, but the choices of data analysis and presentation could be improved. Nevertheless, they are enough to reach the conclusions.

**Specific comments:**

1. PCA analysis: you still have not mentioned how many and which variables were used, and if there was any data preparation for the analysis (such as normalization). The manuscript needs to provide all the information necessary to reproduce the work.

2. PCA results: I personally cannot extract any conclusive information from this section. I understand that the biplot can be done even if the variance explained is only 60%, but I still think this is not the best choice of analysis. The fact that the third PC is almost as important as the second tells me that this biplot is a poor choice, as conclusions could change significantly if the third dimension was included. Considering that there are so many options of statistical analysis, I don't understand why using this one. All the conclusions I mentioned above do not depend on the PCA analysis, so it could be removed.

3. Simulation results: if I understood correctly, you varied the amplitude and the length scale of the heterogeneity resulting in 144 different simulations, but you combined all of them together in your statistics, looking at tower location as the independent variable. This is the reason for the large error bars in Figures 3 and 5, correct? So, why didn't you look into how these two variables (amplitude and length scale) affected your results? Why perform so many simulations? Would the results and conclusions be different if you looked into a smaller set of amplitude and length scale?

4. Is the homogeneous control run also lumped together with everything else in Figures 3-6? Some comparison between homogeneous and heterogeneous cases would be important too.

5. Figures 3 and 5: "That is, we averaged over the data with different time stamps and also over all cases corresponding to the intermediate length scales formed by the choice of surface flux amplitude and length scale." I don't understand this sentence. What intermediate length scales? What choice of surface flux amplitude and length scale? I understood that each data point is a combination of 144 cases, all at the same location independent of the value of surface flux at that location. "The towers are ordered according to the available energy at their location, for our model setup the available energy is equal to the surface flux." you mean "ordered" in the x-axis? Is it the relative or absolute value of the available energy at their location? The use of the "scaled available energy" as the abscissa is to represent the relative location of updraft/downdraft? I am assuming this from the discussion in paragraph 2 of section 3.2, but it is not really clear. Please clarify how the plot was constructed, and justify the choice of variable for the abscissa.

**Suggestions:**

1. Simulation results: I believe it would be useful to see how the variables of the simulation, for example, velocity, temperature, heat flux, advection, flux divergence, look like as a function of height and how do they vary as a function of horizontal directions due to the heterogeneity. I believe that seeing and discussing the physics of the flow and how these variables behave in the spatial domain could help us visualize what is happening physically. Performing LES is a great opportunity to look into these things that we don't have in field experiments. This is a minor suggestion that I believe would help to interpret the statistics better.

2. Equation (3): a sketch of the fluxes and a figure showing where in the 50 by 50m box each term is being calculated would be useful. Another way to avoid copywrite issues is to make a figure yourself. Just a suggestion.

**1    Technical comments:**

- Could you add the ranges of surface flux and $u_*$ in Tables 2 and 3?

---

## Author Response (AR2)

**Reply to the reviewers**

**Frederik De Roo and Matthias Mauder**

**March 2018**

*We thank the reviewers again for their careful inspection of the manuscript and for their remarks and suggestions. We replaced $u^*$ by $u_*$ (because of the source we refer to). As in the previous round, an italic script indicates our response to the reviewer, and Roman script indicates our changes and additions in the manuscript. For clarity our answers are colored in blue, reviewers' comments in black. In red we have highlighted some paragraphs that we removed from the manuscript, as suggested by the reviewer.*

**1 Reviewer 1**

**1.1 Specific comments**

1. PCA analysis: you still have not mentioned how many and which variables were used, and if there was any data preparation for the analysis (such as normalization). The manuscript needs to provide all the information necessary to reproduce the work.

   *We added the names of the variables explicitly (in the methods we had previously written "The correlation matrix contains exactly the same variables that are plotted in the biplot."). Normalization and centering is understood for a correlation biplot, which we mentioned in the methods, but we now repeat it in the results section as well to be as clear as possible. We also repeat it in the figure caption.*

   **In the methods:** The correlation matrix contains exactly the same variables that are plotted in the biplot: besides EBR, we selected the friction velocity (momentum flux) $u^*$, the boundary-layer height $z_i$, the temperature difference between the surface and the tower measurement $T_s - T$, the (total) advection by the mean flow, and the horizontal flux-divergence, with the latter two normalized by the surface flux. This leads to a six-dimensional data space.

   **In the results:** The biplots are based on the PCA explained in section 2.3, and the data variables are first centered and normalized by their standard deviation.

   **In the caption of figure 8:** For the PCA behind these correlation biplot, we only use the six variables shown in the plot. For a correlation biplot the variables are centered around their mean and normalized by their standard deviation. The EBR

is used as the supervised variable. The abscissa is the first principal component (PC1) from the PCA, the ordinate is the unit vector perpendicular to PC1 and lying in the plane spanned by EBR and PC1, i.e. it is the vector given by the difference between EBR and its projection on PC1, including subsequent normalization.

2. PCA results: I personally cannot extract any conclusive information from this section. I understand that the biplot can be done even if the variance explained is only 60%, but I still think this is not the best choice of analysis. The fact that the third PC is almost as important as the second tells me that this biplot is a poor choice, as conclusions could change significantly if the third dimension was included. Considering that there are so many options of statistical analysis, I dont understand why using this one. All the conclusions I mentioned above do not depend on the PCA analysis, so it could be removed.

*We follow the suggestion of the reviewer and we have removed the PCA section and biplot, and we abbreviated the discussion on the correlation analysis.*

**Abridged discussion: (Correlations with the energy balance ratio)**

We investigate the possible connection between the energy balance ratio, the different flux contributions, and variables such as friction velocity and boundary-layer height. We performed a linear correlation analysis between these variables and the energy balance ratio. We made one restriction on the data set, which is to limit the boundary-layer depth to values larger than 1 km, thereby excluding about 8% of the data, in order to obtain a better representation of the boundary-layer depth (when the boundary-layer depths smaller than 1 km are included, the correlation deteriorates).

We found that friction velocity and boundary-layer depth cluster are well-correlated with each other, but not with EBR. Although we might have supposed that higher boundary-layer heights will arise if patches are present with vigorous surface heating, however we found that $u_*$ decreased with stronger surface heterogeneity. Closer analysis reveals that the highest boundary layer heights are obtained when the heterogeneity amplitudes are smaller and the domain is more homogeneous. Hence the former clustering can be explained because in our scenario with varying heterogeneity amplitudes the highest boundary-layer height and larger $u_*$ are both obtained for smaller heterogeneity amplitudes. Though advection and flux-divergence correlate well with EBR, they cannot be measured independently and therefore cannot be used as independent predictors.In the literature (e.g. Stoy et al 2013, Eder et al 2015a) a correlation between friction velocity and energy balance closure has been found: a high friction velocity leads to a smaller residual. Typically, a higher friction velocity is correlated to smaller atmospheric instability, and hence roll-like convection instead of cellular convection. Maronga and Raasch (2013) found that boundary-layer rolls "smear out" the surface heterogeneity, leading to an effective surface that looks less heterogeneous, which has been related to a higher EBR (Mauder et al 2007, Stoy et al 2013). Therefore a possible cause for the present low correlation of $u_*$

with the EBR, could be our range of the stability parameter. For the free convective cases considered here the stability parameter lies below the range where the friction velocity has a high correlation with EBR.

The linear correlation analysis shows that the simulated EBR does not linearly depend on easily measured characteristics. As we have learned from Fig. 5, there can be a good fit between the parameter-averages of two variables, e.g. normalized flux-divergence and energy balance ratio average, despite the fact that the individual data points do not correlate as well. This highlights the importance of testing parameterizations for the energy balance closure problem on the level of a data ensemble, instead of parameterizing on the level of the individual hourly measurements.

**Abstract:** Finally, we seek correlators for the energy balance ratio and the energy residual in the simulations. The correlation with the friction velocity is less pronounced than previously found, but this is likely due to our concentration on effectively strongly to freely convective conditions.

**Conclusions:** We did not find a high correlation between the friction velocity and energy balance ratio, but this could be due to the limited range of $u_*$ as we have investigated free convection.

*Before taking the analysis out, we investigated it more deeply by means of a so-called supervised correlation analysis (and supervised biplot) which concentrates on the exact correlation with a supervised variable (in our case EBR). However, the supervised biplot does not yield additional insights and therefore we exclude it from the revised manuscript.*

**Removed from 2.3**

For investigating the response of the virtual tower measurements to the changes in the parameters, and for investigating the correlation between the measured variables, a principle component analysis (PCA) is applied. PCA relies on the singular value decomposition (SVD) of the data matrix, which consists of the data points for each of the data variables. Through SVD, the data matrix decomposes into the matrix of the left eigenvectors, a diagonal matrix with the singular values and a matrix with the right eigenvectors. The singular values are ordered by their magnitude, because the square of each singular value is the variance of the data explained by its corresponding eigenvectors. Hence the first eigenvectors with the largest singular values represent the principal components that explain the largest fractions of the data variability. We will present many of our results in correlation biplots introduced by Gabriel (1971,1978). Correlation biplots offer a picture of the relationship between the interdependent variables that make up the data matrix through the PCA method. First of all, for a correlation biplot, each variable is centered around its mean and normalized by its standard deviation. On the normalized data matrix PCA is applied. The data variables are projected into the subspace spanned by the first principal components and then the vectors of the projection within this subspace are plotted in a 2D (or a 3D) biplot, when the two (or three) largest principal components are

chosen. These principal components express the (orthogonal) directions in which the data has the most variability. In a correlation biplot, the inner product between the variable vectors (hence, the product of their length and the cosine of the angle) directly measures their correlation. The scree plots related to the biplots express how much variance is captured by the principal components by plotting the (relative) variance explained by the principal components. The variance explained is a measure for the goodness of the fit. The better the variable can be explained by the first two principal components, the longer the length of the vector of the variable in a two-dimensional correlation biplot, which is at most unity, indicated by the unit circle. For a pedagogical description of biplots see e.g. Greenacre (2010).

As the $N$-dimensional biplot shows how well the variables are explained by the first $N$ principal components, standard correlation biplots can be used to investigate the clustering of certain variables: when the arrows of the variables lie close to each other (resp. opposite), their (anti-)correlation is high and consequently these variables predict each other relatively well, so the full data can be described in terms of fewer variables. Simultaneously, biplots can be used to investigate the correlation between two variables, or between one variable of particular interest and the other variables, in which case a supervised biplot is even more adequate. Hence, we have chosen to present two-dimensional correlation biplots with a supervised variable, that is, we plot the projection of the variable vectors into the subspace spanned by the first principal component ($PC1$) and the supervised variable — otherwise, for a standard correlation biplot, the variable vectors are projected in the subspace spanned by the first two principal components. By construction the supervised variable still has unit length in the biplot, as it is already completely in this plane and its length was already normalized during the first step of the PCA analysis. In this supervised biplot, the inner product between the two-dimensional projection of any variable with the supervised variable is the exact expression of the correlation between these the variable and the supervised variable. This exact representation does not hold for correlations among any pair of the variables except for the supervised variable. For these pairs of variables the approximation holds up till the extent that the first principal components capture sufficient variability of the data as shown in the scree plot.

In practice, we are most interested in the energy balance ratio (7) and therefore we choose the EBR as the supervised variable. Its correlation (corr) with another variable is given by their inner product, i.e. the product of the length of the variable arrow and the cosine of the angle of the variable arrow with the EBR arrow in the supervised biplot:

$$\text{corr}(EBR, V) = \frac{\text{Cov}(EBR, V)}{\text{Std}(EBR)\text{Std}(V)} = E\hat{B}R \cdot \hat{V} = \mathbf{EBR} \cdot \mathbf{V} = |\mathbf{V}| \cos v, \quad (1)$$

with $\hat{V}$ the (normalized) variable vector in the full data space of the PCA, $\mathbf{V}$ its projection into the plotted two-dimensional subspace, $|V|$ the arrow length in the

plot and $\upsilon$ the angle in the plotted subspace between the variable and **EBR**. Due to the choice of EBR as the supervised variable and the construction of the supervised biplot in the subspace spanned by $PC1$ and **EBR** it follows that

$$\mathbf{EBR} = \mathbf{E\hat{B}R}\,. \tag{2}$$

For producing the biplots we have made use of Python3, combining our own routines with standard packages. The correlation matrix contains exactly the same variables that are plotted in the biplot: besides EBR, we selected the friction velocity (momentum flux) $u_*$, the boundary-layer height $z_i$, the temperature difference between the surface and the tower measurement $T_s - T$, the (total) advection by the mean flow, and the horizontal flux-divergence, with the latter two normalized by the surface flux. This leads to a six-dimensional data space.

**Removed from 3.4:**

We now turn our attention to the possible connection between the energy balance ratio, the different flux contributions, and other measured variables like the friction velocity. We investigate the correlations for the data sets of Tables 2–3 in separate correlation biplots (Fig. 8) with the EBR as supervised variable. The biplots are based on the PCA explained in section 2.3, and the data variables are first centered and normalized by their standard deviation. We made one restriction on the data set, which is to limit the boundary-layer depth to values larger than 1 km, thereby excluding about 8% of the data, in order to obtain a better representation of the boundary-layer depth within the biplots (when the boundary-layer depths smaller than 1 km are included, the correlation deteriorates). Within the subspace spanned by the first two principal components, the temperature difference between surface and measurement height has the highest correlation to the EBR, however, for the full 6D variable space there is not such a good correlation between EBR and temperature difference, as can be seen in the supervised correlation biplot, where the angle between the EBR and temperature difference tends towards 90 degrees. The temperature difference between the surface and the measurement height can be considered as proportional to the temperature gradient in the surface layer. Correlating EBR with the temperature gradient from the surface with respect to the center of the mixed layer did not improve the correlation.

From the scree plots in Fig. 9 that depict the fraction of explained variance, it can be seen that there is still some scatter in the data and the explanatory strength of the two most important principal components is not very large (a combined 61% and 63% respectively). This is also the reason why we have limited the number of variables that are highlighted in the biplots to only 6 (the ones shown in the biplot) as more variables always introduced additional variance, and the explanatory power of the most important principal components decreased further. The third component has almost the same fraction as the second component. For the supervised biplot, the supervised variable mainly lies along in PC3 with a fraction of PC2 as well.

Despite the variance explained of around 60%, plotting the correlation biplot is still possible, it remains a graphical representation of the correlation between the variable vectors, even though it shows that the variance in those variables cannot be simply explained by the first two principal components. In addition, the supervised correlation biplot shows the correlations with EBR exactly, independent of the explained variance. Furthermore, the linear correlation analysis in the biplots is still useful in a few other respects: for one, it shows that EBR does not linearly depend on easily measured characteristics. For another, the biplots represent the correlation between the individual (hourly) data points, whereas the regression lines in Figs. 5 and 7 were obtained from averaging those individual data points over similar tower locations. It is somewhat puzzling that the individual hourly data points leading to the biplots do not show such a high correlation.

**Removed from the conclusions**

However, we found an unexpected correlation between the energy balance ratio and the difference between sonic temperature and surface temperature. Both can be measured readily by typical micro-meteorological stations. Therefore, this difference could be a promising predictor for a potential energy balance closure correction. Nevertheless, the virtual measurement height remains a critical issue and further investigations are needed with more realistic measurements heights to confirm such a relationship.

[Figure]

Supervised correlation biplots for the kilometer scale heterogeneity (left panel) and the hectometer scale heterogeneity (right panel) with EBR as the supervised variable. For the PCA behind these correlation biplots, we only use the six variables shown in the plot. For a correlation biplot the variables are centered around their mean and normalized by their standard deviation. The EBR is used as the supervised variable. The abscissa is the first principal component (PC1) from the PCA, the ordinate is the unit vector perpendicular to PC1 and lying in the plane spanned by EBR and PC1, i.e. it is the vector given by the difference between EBR and its projection on PC1, including subsequent normalization.

[Figure]

Scree plots of the biplot analyses of Fig. 8 with all 6 principal components. Left panel: order kilometer length scale. Right panel: order hectometer length scale. As far as the explained variance is concerned, both orders of length scale behave very similarly. The fraction of the explained variance that is visible in the supervised biplot is colored in yellow. For a standard two-dimensional biplot, only PC1 and PC2 would contribute to the explained variance in the biplot, but in the supervised biplot we project onto the plane in which the supervised variable lies. In this case the supervised variable is mainly aligned along PC3 with some contribution from PC2 (and PC4 for the hectometer scale), and these fractions build the explained variance of the supervised biplot in addition to PC1.

3. Simulation results: if I understood correctly, you varied the amplitude and the length scale of the heterogeneity resulting in 144 different simulations, but you combined all of them together in your statistics, looking at tower location as the independent variable. This is the reason for the large error bars in Figures 3 and 5, correct? So, why didn't you look into how these two variables (amplitude and length scale) affected your results? Why perform so many simulations? Would the results and conclusions be different if you looked into a smaller set of amplitude and length scale?

*We investigated the amplitude variations in more detail as well, which was the original reason for the size of the set of amplitude variations. The EBR (and other statistics) didn't show a clear pattern in function of the amplitudes ($A_x$ and $A_y$) and therefore we didn't include it in the article. We now add a comment. As to the size of the set, if we drop amplitudes $0.2$ and $0.4$ (thinning the set without changing its range) the results are indeed very similar, some data points shift maybe $1 - 2\%$ in value, but nothing qualitatively different. So with hindsight we could indeed have done with a few simulations less. Nevertheless for analyzing the statistics for the suite of one length scale, since we had these simulations available, we use their variation (in the simulations with different amplitudes) as well.*

**Added in 3.2.:** We also analyzed the variation of EBR in function of the surface amplitudes ($A_x$ and $A_y$) but didn't find any clear dependence there.

*We focus on the length scale's order of magnitude, not on the exact length. As the flow is inherently three-dimensional and the surface two-dimensional, the length scale of the surface pattern cannot be exactly captured with a single number (a surface with discs would behave slightly differently from a surface with squares as we have here) so we concentrate on the order of magnitude.*

**Added in 2.1.:** One suite is focused on kilometer scale heterogeneity, the other on hectometer scale heterogeneity. As the surface heterogeneity is two-dimensional, the length scale of the surface pattern cannot be exactly captured by a single numbers and therefore we concentrate on the order of magnitude of the length scale, and not on the exact length, thus comprising 4 combinations of length scales ($L_x$ and $L_y$) within the suite of kilometer scale (resp. hectometer scale) heterogeneity.

*The error bars indeed show the spread. We had written in the article: "The error bars on the normalized fluxes denote the spread on the virtual measurements of each tower with respect to the suite. The spread is naturally quite large since at each tower, different amplitudes for the surface heat flux pattern are considered."*

4. Is the homogeneous control run also lumped together with everything else in Figures 3-6? Some comparison between homogeneous and heterogeneous cases would be important too.

*We had actually lumped it in because we have a case that satisfies $A_x = A_y = 0$ which could be considered as the "limit of vanishing surface heterogeneity". However, one*

*could equally well argue that it should be left out. As we will now introduce a direct comparison with the homogeneous results, we decided to update the heterogeneous plots as well, by taking this case with $A_x = A_y = 0$ out. Our results are practically unchanged, as this only forms a small fraction (1/36) of the suite, so we do not need to update the text.*

**In 3.2.:** That is, we averaged over the data with different time stamps and also over all cases within the suite corresponding to the kilometer length scale: this entails $(6 \times 6 - 1)$ variations of the surface flux amplitudes (we do not count the case where both amplitudes are zero, $A_x = A_y = 0$, as this is a homogeneous run) multiplied by $2 \times 2$ variations of the heterogeneity length as expressed in table 2.

*In order to make a direct comparison between the suite and the homogeneous runs, in Figs. 4 and 6 we have added an additional panel for the homogeneous runs. These include 4 runs with each 9 towers, though all of those are naturally in the same environment. These 36 cases for the homogeneous average are then better comparable with the 35 amplitude variations for each of the heterogeneous towers (though the latter have another factor of 4 length scale variations). We changed the marker for the tower in the coolest patch into a pentagon because we wanted to reserve the circle for the homogeneous runs (we updated all Figures involved).*

**Added in 3.2:**

In the right panel we show the data from four homogeneous control runs (with data extraction window and data selection in the same manner as for the heterogeneous runs). Each of these simulations has nine towers as well, but now all towers have the same surface properties. The mean residual (underclosure) of the homogeneous control runs is around 10%, less than for the heterogeneous cases but not negligible. There is significant spread on the results, but the residual is mainly composed of advection and storage. Compared to the towers at the edges (middle panel), which are locally heterogeneous, the homogeneous case is clearly different. Compared to the towers at the centers of the patches (left panel) the homogeneous case has a different average but the difference is still within the spread. It is remarkable that flux-divergence is very small in the homogeneous case, in contrast to the heterogeneous terrain. The negligible flux-divergence for a homogeneous site was also apparent in the desert site of Eder et al (2015).

[Figure]

Control volume fluxes as a function of available energy (scaled by the median value) for kilometer scale landscape heterogeneity. The fluxes are normalized by the available energy at their respective location, in our setup this means normalization by the surface flux. Please note that we have plotted the non-closure (normalized energy balance residual) instead of the energy balance ratio EBR (normalized turbulent flux). The left panel shows the towers at the centers of the patches, the middle panel the towers at the edges of the patches, and the right panel the results for the homogeneous control runs. For the tower symbols, see Fig. 2. The error bars denote the spread over the different cases of surface heterogeneity within the suite of kilometer scale surface heterogeneity. The abscissa is the available energy at the tower, but scaled by the mean available energy of the nine towers for that case. In this way, we can group the towers by tower type, also for the cases with different surface amplitudes. Thus, the low values represent the towers located at the cooler patches (downdrafts), the high values the towers located at the hotter patches (updrafts). See text for further discussion.

[Figure]

Control volume fluxes as a function of available energy (scaled by the median value) for hectometer scale landscape heterogeneity. The fluxes are normalized by the available energy at their respective location, in our setup this means normalization by the surface flux. Please note we have plotted the non-closure (normalized energy balance residual) instead of the energy balance ratio EBR (normalized turbulent flux). The left panel shows the towers at the centers of the patches, the middle panel the towers at the edges of the patches, and the right panel the results for the homogeneous control runs. For the tower symbols, see Fig. 2. The error bars denote the spread over the different cases of surface heterogeneity within the suite of hectometer scale surface heterogeneity. The abscissa is the available energy at the tower, but scaled by the mean available energy of the nine towers for that case. In this way, we can group the towers by tower type, also for the cases with different surface amplitudes. Thus, the low values represent the towers located at the cooler patches (downdrafts), the high values the towers located at the hotter patches (updrafts). See text for further discussion.

**Added in 3.3:**

Again, in the right panel we show the data from four homogeneous control runs. However, except for the flux-divergence, the tower responses in heterogeneous terrain of hectometer scale heterogeneity look similar to the homogeneous runs.

5. Figures 3 and 5: "That is, we averaged over the data with different time stamps and also over all cases corresponding to the intermediate length scales formed by the choice of surface flux amplitude and length scale." I don't understand this sentence. What intermediate length scales? What choice of surface flux amplitude and length scale? I understood that each data point is a combination of 144 cases, all at the same location independent of the value of surface flux at that location. "The towers are ordered according to the available energy at their location, for our model setup the available energy is equal to the surface flux." you mean "ordered" in the x-axis? Is it the relative or absolute value of the available energy at their location? The use of the "scaled available energy" as the abscissa is to represent the relative location of updraft/downdraft? I am assuming this from the discussion in paragraph 2 of section 3.2, but it is not really clear. Please clarify how the plot was constructed, and justify the choice of variable for the abscissa.

*In retrospect, we agree that our formulation isn't easy to read. We've rewritten the figure caption. With intermediate length scales we meant, for example, that we have heterogeneity lengths of 1.5 and 3 kilometers within the suite of kilometer scale heterogeneity.*

**In 3.2.:**That is, we averaged over the data with different time stamps and also over all cases within the suite corresponding to the kilometer length scale: this entails $(6 \times 6 - 1)$ variations of the surface flux amplitudes (we do not count the homogeneous case $A_x = A_y = 0$) multiplied by $2 \times 2$ variations of the heterogeneity length as expressed in table 2.

**Added in the figure caption:** The left panel shows the towers at the centers of the patches, the middle panel the towers at the edges of the patches, and the right panel the results for the homogeneous control runs. For the tower symbols, see Fig. 2. The error bars denote the spread over the different cases of surface heterogeneity within the suite of hectometer scale surface heterogeneity.

*As we want to group towers of the same tower type together, we have to take into account the cases with different surface amplitudes. For this reason we scale the available energy at the tower by the mean available energy of the towers for that case. This maps towers of the same tower type at the same location of the abscissa. We added this to the figure captions.*

**Added in the figure caption:** . The abscissa is the available energy at the tower, but scaled by the mean available energy of the nine towers for that case. In this way, we can group the towers by tower type, also for the cases with different surface

amplitudes. Thus, the low values represent the towers located at the cooler patches (downdrafts), the high values the towers located at the hotter patches (updrafts).

**1.2  Suggestions**

1. Simulation results: I believe it would be useful to see how the variables of the simulation, for example, velocity, temperature, heat flux, advection, flux divergence, look like as a function of height and how do they vary as a function of horizontal directions due to the heterogeneity. I believe that seeing and discussing the physics of the flow and how these variables behave in the spatial domain could help us visualize what is happening physically. Performing LES is a great opportunity to look into these things that we dont have in field experiments. This is a minor suggestion that I believe would help to interpret the statistics better.

*The turbulent flux, advection and flux-divergence were in this suite only calculated for the designed control volumes so unfortunately we cannot show their horizontal and vertical dependence here (they depend on spatial and temporal correlations and these are expensive to store). We included a horizontal cross-section for the velocity in section 3.1 for some specific simulations, which gives the basic structure of the flow.*

2. Equation (3): a sketch of the fluxes and a figure showing where in the 50 by 50m box each term is being calculated would be useful. Another way to avoid copywrite issues is to make a figure yourself. Just a suggestion.

*We added a sketch of the fluxes in the control volume (Fig. 1):*

[Figure]

Graphical representation of (1). The control volume is colored in yellow, with horizontal flux-divergence in green, the advection terms in blue, and the storage flux in cyan. The surface flux and the measured turbulent flux are both in black. For clarity the lateral dimension perpendicular to the cross-section is not shown. The direction of the arrows indicate a positive contribution.

**1.3 Technical remarks**

- Could you add the ranges of surface flux and u* in Tables 2 and 3?

*Ranges have been added.*

**2 Reviewer 2**

The authors addressed many of my concerns, including a correction of an equation. However, the second read of the manuscript was not satisfying and I will try to give a constructive reasoning below.

**2.1 Comments on the author's reply to the first revision**

1. Major comment 1. I commented that the authors should be careful when integrating of large volumes (here 50m height) and not considering the logarithmic temperature profile. They answered that the logarithmic profile only used as boundary condition and thus affects the region from 0 to 5m above ground. I disagree on this point. The logarithmic profile is supposed to be valid in the entire surface layer. With a boundary layer height of 1 km as in the present simulations, the surface layer extents up to say 100 m, whatever it assumed as boundary condition. Please show me the temperature profile between 0 m and 50 m to prove that its shape can be approximated by a linear function.

   - *We apologize for our misunderstanding. The logarithmic profile obtained from MOST is only used as the boundary condition. This is indeed a different topic than the shape of the potential temperature in the surface layer.*

   - *Nevertheless, we do not need to assume that the shape of the potential temperature in the surface layer can be approximated by a linear function (we expand on this below).*

   - *Furthermore, in an implicit LES the numerical integration over the grid boxes is straightforward. We explain the method that we use below, and we have added a paragraph to the methods section to make it clear for the reader as well.*

   Going into more details, we want to clarify our integration method:

   (a) *With "approximated by a linear function", we presume that the reviewer is referring to the application of the trapezoidal rule in numerical integration, where for each sampling interval (in our case each grid cell) of the total numerical integration interval, the function is approached by a piecewise linear function. When the approximated function does not vary wildly within the sampling interval, which is the case for a logarithm (this condition does not have to be satisfied for the entire integration interval at once) errors cancel each other out to a large extent and become smaller when the sampling interval is made smaller.*

   (b) *We actually make use of the "midpoint rule" (interpolation function is piecewise constant) and we have 5 grid points up to 50 meters. Again, numerical errors cancel each other out (partly) and disappear when the sampling intervals are made small. How small the interval has to be, depends on the behaviour of the*

*function that is approximated. For the storage term, only the resolved potential temperature has to be used (there is no sub-grid contribution here, they exist only for the fluxes in the LES). Furthermore the time change over 1 hour is averaged out again, which makes it less susceptible to turbulent motion. We only need to make assumptions on the regularity of the potential temperature. Of course, the numerical integration error will be smaller in the case we can assume the shape is closer to piecewise constant, but we do not need to assume a linear shape over the whole integration range in order to apply numerical midpoint integration and we can certainly interpolate the logarithm between $z_0$ and 50 meters on five grid levels, especially because a logarithm for large values of the argument $(z/z_0)$ is hardly changing and hence for integration purposes the piecewise constant approximation is really a good approximation for large values of the argument $(z \gg z_0 = 0.1 \, \mathrm{m})$, because $\lim_{x \to \infty} \frac{d}{dx} \log(x) \to 0$. Furthermore, any remaining numerical error can only be a fraction of an already small term (the storage term does not contribute significantly). Hence we only need to assume that the function can be approximated by a piecewise constant function over each of the sampling intervals (this is considerably more flexible than the linear function over the whole integration interval that the reviewer is referring to).*

(c) *In an LES such as PALM that uses "implicit filtering", the representation of the resolved quantities has a so-called "finite volume" character: it is by construction assumed that the scalar in the grid box is already the spatial mean of the grid box, therefore the midpoint rule (which otherwise assumes piecewise constant interpolation function) is the most appropriate, because the LES computed $\theta[k]$ is not $\theta(z = z_k)$ but already $\int_{z_k-dz}^{z_k+dz} \theta(z)dz$, with $z_k$ the height of the grid point $k$, $dz$ the grid spacing and $\theta$ the potential temperature. Therefore, to remain self-consistent with the LES, midpoint integration is the natural choice, and yields the desired result for an implicit LES.*

**In the methods:**

For the integration of the temperature in the storage term we apply numerical integration with the midpoint rule, which assumes a piecewise constant interpolation function. PALM uses implicit filtering, where it is by construction assumed that the prognostic variable within the grid cell is the volumetric mean of the variable over the domain of the grid cell, therefore the midpoint rule is the most appropriate, because by definition the LES computed $\theta[k]$ is not $\theta(z = z_k)$ but instead

$$\theta[k] = \int_{z_k-dz}^{z_k+dz} \theta(z)dz \, , \tag{3}$$

with $z_k$ the height of the grid point $k$, $dz$ the grid spacing and $\theta$ the potential temperature, and we have suppressed the indices $ij$ for clarity. In this way, the summation of the LES computed discrete profile values is defined to be equal to the

integration of the continuous profile,

$$\sum_{k=1}^{K} \theta[k] = \int_{0}^{z_m} \theta(z)dz \,, \tag{4}$$

with the measurement height $z_m = z_K + dz$.

2. Major comment 2. The authors state that they performed homogeneous control runs, but actually, I do not see any data analysis for these runs. Hence, my comment is still not addressed. I still want to see that the residual is zero for this case! And if not, it might indicate that the time-averaging was not long enough (see below).

*We added the homogeneous control runs to Figs. 4 and 6 and added some lines to the discussion. We are working along the same lines as the LES literature, as previous studies have found non-zero residuals for homogeneous terrain as well, as mentioned in our introduction (Kanda et al. 2004, Steinfeld et al. 2007, Huang et al., 2008). These studies showed that turbulent organized structures develop also over an ideal homogeneous surface. They are often quasi-stationary and do not propagate with the mean wind, i.e. they violate the ergodic hypothesis. Therefore, the transport associated with these can inherently not be captured by measurement at a certain point. Only a spatial covariance would be able to do so.*

**Added in 3.1.:**

Due to the absence of a background wind, significant circulation patterns can emerge in the homogeneous case as well. With even longer averaging times a zero mean can be achieved for idealized simulations in homogeneous terrain, but in a real atmospheric boundary-layer this is not possible due to non-stationarity on those timescales.

**Added in 3.2:** In the right panel, we show the data from four homogeneous control runs (with data extraction window and data selection in the same manner as for the heterogeneous runs). Each of these simulations has nine towers as well, but now all towers have the same surface properties. The mean residual (underclosure) of the homogeneous control runs is around 10%, less than for the heterogeneous cases but not negligible. There is significant spread on the results, but the residual is mainly composed of advection and storage. Compared to the towers at the edges (middle panel), which are locally heterogeneous, the homogeneous case is clearly different. Compared to the towers at the centers of the patches (left panel) the homogeneous case has a different average but the difference is still within the spread. It is remarkable that flux-divergence is very small in the homogeneous case, in contrast to the heterogeneous terrain. The negligible flux-divergence for a homogeneous site was also apparent in the desert site of Eder et al (2015).

**Updated figures:**

[Figure]

Control volume fluxes as a function of available energy (scaled by the median value) for kilometer scale landscape heterogeneity. The fluxes are normalized by the available energy at their respective location, in our setup this means normalization by the surface flux. Please note that we have plotted the non-closure (normalized energy balance residual) instead of the energy balance ratio EBR (normalized turbulent flux). The left panel shows the towers at the centers of the patches, the middle panel the towers at the edges of the patches, and the right panel the results for the homogeneous control runs. For the tower symbols, see Fig. 1. The error bars denote the spread over the different cases of surface heterogeneity within the suite of kilometer scale surface heterogeneity. The abscissa is the available energy at the tower, but scaled by the mean available energy of the nine towers for that case. In this way, we can group the towers by tower type, also for the cases with different surface amplitudes. Thus, the low values represent the towers located at the cooler patches (downdrafts), the high values the towers located at the hotter patches (updrafts). See text for further discussion.

[Figure]

Control volume fluxes as a function of available energy (scaled by the median value) for hectometer scale landscape heterogeneity. The fluxes are normalized by the available energy at their respective location, in our setup this means normalization by the surface flux. Please note we have plotted the non-closure (normalized energy balance residual) instead of the energy balance ratio EBR (normalized turbulent flux). The left panel shows the towers at the centers of the patches, the middle panel the towers at the edges of the patches, and the right panel the results for the homogeneous control runs. For the tower symbols, see Fig. 1. The error bars denote the spread over the different cases of surface heterogeneity within the suite of hectometer scale surface heterogeneity. The abscissa is the available energy at the tower, but scaled by the mean available energy of the nine towers for that case. In this way, we can group the towers by tower type, also for the cases with different surface amplitudes. Thus, the low values represent the towers located at the cooler patches (downdrafts), the high values the towers located at the hotter patches (updrafts). See text for further discussion.

**Added in 3.3:**

Again, in the right panel we show the data from four homogeneous control runs. However, except for the flux-divergence, the tower responses in heterogeneous terrain of hectometer scale heterogeneity look similar to the homogeneous runs.

3. Minor comment 6 and 14. The authors state in their answer that the heterogeneity creates a non-zero friction velocity. Now, strictly speaking the friction velocity is a parameter based on MOST and which does not (strictly speaking) exist when the flow is heterogeneous. I am aware that it is used nevertheless and in LES models it is even used LOCALLY (another violation of MOST). Unfortunately, I have the feeling that the authors use these terms rather loosely and I would be happy to have a more precise use and/or additional comments in the manuscript. In summary: friction velocity is only defined properly when a) the flow is homogeneous and b) when it is defined based on a mean profile. It is zero for free convection. And yes, when it is measured it is ¿ 0 because of the local shear created by thermals, but that is partly because of the violations explained above.

*We consider the friction velocity as obtained through the momentum flux (velocity covariances). We have incorporated the remarks of the reviewer into the manuscript and clarified our usage of friction velocity:*

**In section 1.3:** as we derive the friction velocity from the kinematic momentum flux $(\tau_0/\rho)$, in the same manner as how it is applied in standard eddy-covariance measurements (e.g. Kaimal and Finnigan, 1994):

$$u_*^2 = \tau_0/\rho = \left( \overline{u'w'}^2 + \overline{v'w'}^2 \right)^{1/2} .$$ (5)

This definition of friction velocity by the momentum flux is found in general fluid mechanics as well (e.g. Landau and Lifschitz 1959). However, only in homogeneous flow, the friction velocity makes sense as a scaling parameter in Monin-Obukhov similarity theory. Therefore, we want to stress that when the friction velocity is derived from the mean velocity gradient, this is only valid in homogeneous flow. For conditions of free convection in homogeneous terrain the friction velocity derived from the mean velocity is clearly zero (even though free convection flow is locally inhomogeneous). As we do not have homogeneous flow in our study of heterogeneous terrain, we will make use of the momentum flux (5) to derive the friction velocity.

*We also add the comment about the lower boundary in an LES:*

**In the methods:** It is worth to note that the application of MOST at the first grid point in an LES, is done locally and based on the instantaneous velocity.

4. Language 10. I still do not like the sentence "we plot" because the plot was already made. You do not plot something while I am reading your manuscript. You can say "we show" or something similar instead.

*All instances replaced by "we show".*

**2.2 Comments on the revised manuscript**

1. Eq. 1 and 2: "H" is used in both with different meaning. Even worse, in Eq. 2 it is not explained that is shall be the heat flux, so the reader is forced to believe it to be the Heaviside function.

   *We agree that we should avoid the confusion. For clarity, we renamed the Heaviside function $\mathcal{H}$ and we added "surface heat flux $H$" explicitly, in order not to have to rely on the context.*

2. P8 L19. So at which height is the flow sufficiently resolved? You use 50 m without justification.

   *The 50 m height is relative to the grid size. We request that the ratio of the subgrid-scale flux to the total flux at that height to be less than $1\%$.*

   **Added in the methods:** Demanding that the subgrid-scale flux does not exceed $1\%$ of the resolved flux, we place our virtual flux measurement at $50\,\mathrm{m}$ height.

3. Fig. 2: This data is at 50 m height, correct? This information is missing in the caption. Then, I am very much surprised that the homogeneous control run displays horizontal variations that large (0.5 m/s is not negligible). You report that you averaged over 4 h of time and ideally one expects zero variance in the resulting fields. The fact that there is so much variance suggests that the averaging time is way from being sufficiently long. This then affects all other runs as well and questions the results in general.

   *The homogeneous run exhibits turbulent organized structures, which are especially apparent because of the absence of the background wind (they become less pronounced with background wind because the circulation pattern is smeared out, even when rolls don't appear). This is in accordance with literature results for unstable flows over a homogeneous surface (e.g. Kanda et al 2004). At the level an individual realization, the LES predicts that there are still turbulence structures in the homogeneous case for free convection. One can indeed remove structures in the simulations by longer averaging times or by considering ensemble runs. For a true ABL with diurnal evolution, the longer averaging time is however not a possibility and in reality ensemble runs are out of the question too. We do not agree that this affects the heterogeneous simulations because there the circulation is expected to persist due to the heterogeneity. Furthermore, because the averaging over the different cases ($35 \times 4$) within each suite can be considered as a very large ensemble run, the additional averaging over the suite removes this type of random turbulence.*

   **Added in the figure caption:** ($z = 50\,\mathrm{m}$)

4. P13 L 28: "one magnitude smaller" than what?

   One magnitude smaller than the length scale considered in the preceding sentence (there the kilometer scale is mentioned).

**Changed into:** "For surface heterogeneity of hectometer scale".

5. General: you state you performed many simulations (288 in total), but I gather that the reader is only confronted with data from six runs. Is this correct or did I understand something wrong here? I was expecting at least some analysis for all runs otherwise you should drop them.

   *This is a misunderstanding, we use all the simulations and consider the average over the suite, in order to find out the average effect of surface heterogeneity of a certain scale. We now stress it even more explicitly.*

   **In 3.2:** This is the average of the simulation output belonging to the suite of kilometer scale heterogeneity. In this manner, we investigate the average effect of surface heterogeneity of kilometer scale.

6. Again, I do not understand the biplots. You make some effort to describe them but I would have to read other literature to understand them. I do not think that this is educationally good. Maybe there are more capable readers than me, but if you want to reach a larger audience why not include a very short and easy tutorial on how to read these plots.

   *On suggestion of reviewer 1, we have removed the section about the PCA and the biplots altogether. We only include a section in the discussion with the main findings of this linear correlation analysis without referring to biplots.*

7. General: Maybe it's just me, but I find the manuscript hard to read. That is also partly because of all the terms like "energy balance ratio" and "available energy (scaled)". Why not use the formula terms in the plots instead?! It would make reading so much easier.

   *We believe that energy balance ratio is a better quantity to plot than the turbulent flux, because the former is normalized (and made dimensionless) by the surface flux, and allows better comparison because the different patches have different surface fluxes. For more clarity we already prepare the reader in the methods in addition to our explanation in the results. Moreover, energy balance ratio is common terminology in the literature about the energy balance closure problem (e.g. Stoy et al., 2013 etc.)*

[revised manuscript text omitted]